# Scalable Event Cloud Network for Event-based Classification

**Hongwei Ren** [* 1] **Fei Ma** [* 2] **Xiaopeng Lin** [3] **Yuetong Fang** [3] **Hongxiang Huang** [3] **Yue Zhou** [3] **Yulong Huang** [3] **Haotian Fu** [3] **Ziyi Yang** [3] **Youxin Jiang** [1] **Xiangqian Wu** [1] **Bojun Cheng** [3]

## Abstract

Event cameras are biologically inspired sensors garnering significant attention from both industry and academia. Mainstream methods favor frame and voxel representations, which reach a satisfactory performance while introducing time-consuming transformations, bulky models, and sacrificing fine-grained temporal information. Alternatively, Point Cloud representation demonstrates promise in addressing the mentioned weaknesses, but it has limited scalability in abstracting features of higher spatial resolution and longer temporal sequence events. In this paper, we propose a **S**calable **N**etwork named SECNet to leverage **E**vent **C**loud representation. SECNet integrates polarity at the structural level by innovating the Event-based Group and Sampling module rather than only at the input level. To accommodate the surge in the number of events, SECNet embraces feature extraction in the frequency domain via the Fourier transform. This approach not only substantially extinguishes the explosion of Multiply Accumulate Operations but also effectively abstracts spatio-temporal features. We conducted extensive experiments on **ten** event-based datasets, and substantiate the scalability, effectiveness, and efficiency of SEC-Net. Our code will be available at: https://github.com/rhwxmx/SECNet_ICML.

## 1. Introduction

Event cameras represent a revolutionary advancement in the field of computer vision (Lichtsteiner et al., 2008; Gallego

*Equal contribution [1]Research Centre for Multimodal Artificial Intelligence and Applications, Faculty of Computing, Harbin Institute of Technology [2]Guangdong Laboratory of Artificial Intelligence and Digital Economy (SZ) [3]MICS Thrust, Hong Kong University of Science and Technology (Guangzhou). Correspondence to: Bojun Cheng <bocheng@hkust-gz.edu.cn>.

*Proceedings of the 43$^{rd}$ International Conference on Machine Learning*, Seoul, South Korea. PMLR 306, 2026. Copyright 2026 by the author(s).

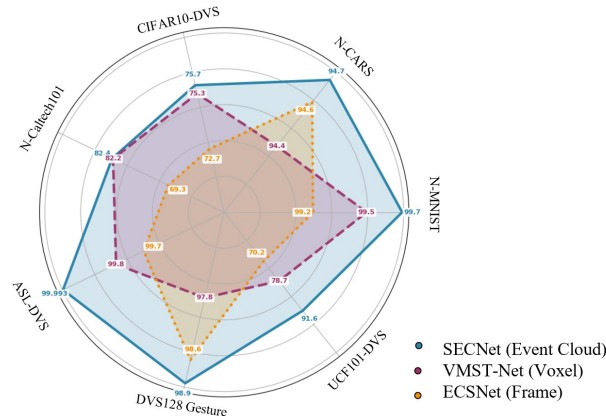

*Figure 1.* The comparison with the SOTA voxel-based VMST-Net (Liu et al., 2023) and frame-based methods ECSNet (Chen et al., 2022b) on seven classification datasets.

et al., 2020). Unlike traditional frame-based cameras that capture images at fixed intervals, event cameras operate asynchronously, detecting local changes in illumination that exceed a predetermined threshold and generating events independently at the pixel level (Posch et al., 2010). This fundamental difference allows them to achieve exceptional performance characterized by high dynamic range, low latency, and low power consumption (Mueggler et al., 2017). Consequently, event cameras demonstrate superior effectiveness in capturing high-speed motion and rapid scene dynamics, generating outputs in microsecond resolution, which far surpasses the frame rate of conventional cameras (Delbruck & Lang, 2013).

The raw events generated by event cameras are represented by four dimensions, 2S-1T-1P coordinates, which comprise 2-Spatial coordinate $(x, y)$, 1-Timestamp $t$, and 1-Polarity $p$ of each event occurrence (Wang et al., 2019a). Due to the boom in deep learning algorithms in computer vision, most efforts in event-based vision primarily focus on converting raw events into frame-based or voxel-based representations, employing established backbones to accomplish various tasks, such as VGG (Simonyan & Zisserman, 2014), ResNet (He et al., 2016), and Transformer (Dosovitskiy, 2020). Although these methods generally yield satisfactory performance, they often introduce additional representa-

tion transformation times and involve relatively large model architectures, which can substantially diminish their compatibility with hardware and applications (Sekikawa et al., 2019), especially for high-resolution scenes as demonstrated in Table 4. On the other hand, the sparseness and fine-grained temporal information contained in raw events are greatly discounted when converted into frames and voxels, which is detrimental for tasks like action recognition that rely on critical details in the temporal dimension (Ren et al., 2025).

Alternative event representation has also been investigated and is gaining popularity. Inspired by PointNet (Qi et al., 2017a) and PointNet++ (Qi et al., 2017b), several studies employ Point Cloud to represent raw events. Such representation exempts the data format transformation process, preserves the fine-grained temporal information, and is theoretically coupled with lighter network architectures. However, there are **three major limitations** preventing Point Cloud-based methods from achieving comparable performance to Frame-based and Voxel-based methods. Firstly, Point Cloud representation often ignores the polarity information or merely treats it as part of the input, while treats the temporal coordinate $t$ as a quasi-spatial dimension $z$ (Wang et al., 2019a; Ren et al., 2023; Sekikawa et al., 2019), failing to differentiate each dimension of events. Secondly, the Point Cloud networks show limited scalability due to insufficient extraction of spatial-temporal features in long sequence events from both space and time. Thirdly, the networks' computational load, which is quantified by the Multiply Accumulate Operations (MACs), super-linearly grows with the increase in input points, leading to potential inefficiencies when the input becomes larger (Chen et al., 2022a); details are provided in Section E.

In order to address the constraints mentioned above, we embrace Event Cloud representation (Wang et al., 2019a), which is the nearest representation to raw events and requires only downsampling. Compared to the previous Point Cloud method, it gives the polarity and includes multiple times the number of events (1024 to 10240). Then, we innovate a scalable, efficient, and effective framework named SECNet by following the steps. Firstly, we redesign the Group and Sampling (G & S) module to actively incorporate polarity not just at the input level (Xie et al., 2022; Liu et al., 2023), but also into structural decisions such as neighborhood formation and coordinate updates. Secondly, we leverage the technique of frequency domain analysis to efficiently extract features in the spatial and temporal domains. The Spatial Frequency-aware (FA) module significantly reduces the MACs by replacing the convolution with Hadamard's product by a repetition of $10^4$. Meanwhile, the Temporal-FA module excels at capturing long-sequence dependencies of Event Cloud by global frequency filters rather than providing local temporal information. Finally, SECNet

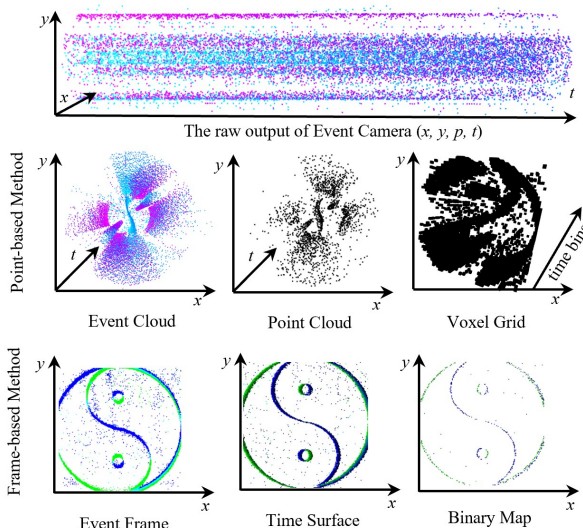

*Figure 2.* Visualization of different representations in yin-yang from N-Caltech101 dataset.

can proficiently capture the spatio-temporal features with a lightweight architecture scale to higher spatial and longer temporal scenes. We conducted extensive experiments on ten datasets for three different tasks: object recognition, action recognition, and human pose estimation. The experimental results sufficiently demonstrate the superiority of SECNet, and the gain in seven datasets is shown in Figure 1. Our main contribution can be summarized in the following:

- SECNet structurally integrates polarity from Event Cloud representation rather than only input level.

- SECNet effectively captures spatio-temporal features from long-term events via frequency domain analysis.

- SECNet is extremely lightweight and scalable to handle high-resolution and long-sequence scenes.

- Comparable results highlight that SECNet serves as a powerful backbone for the community.

## 2. Related Work

### 2.1. Event Representations

Event cameras capture changes in the environment rather than entire frames at regular intervals, allowing them to record rapid movements with high temporal resolution and perform well in challenging lighting conditions. The representation of output from event cameras is a well-developed research problem (Gallego et al., 2020), and we visualize some frame-based and point-based representations in Figure 2. The most common one is frame-based representations, which involve converting event streams into 2D frames by

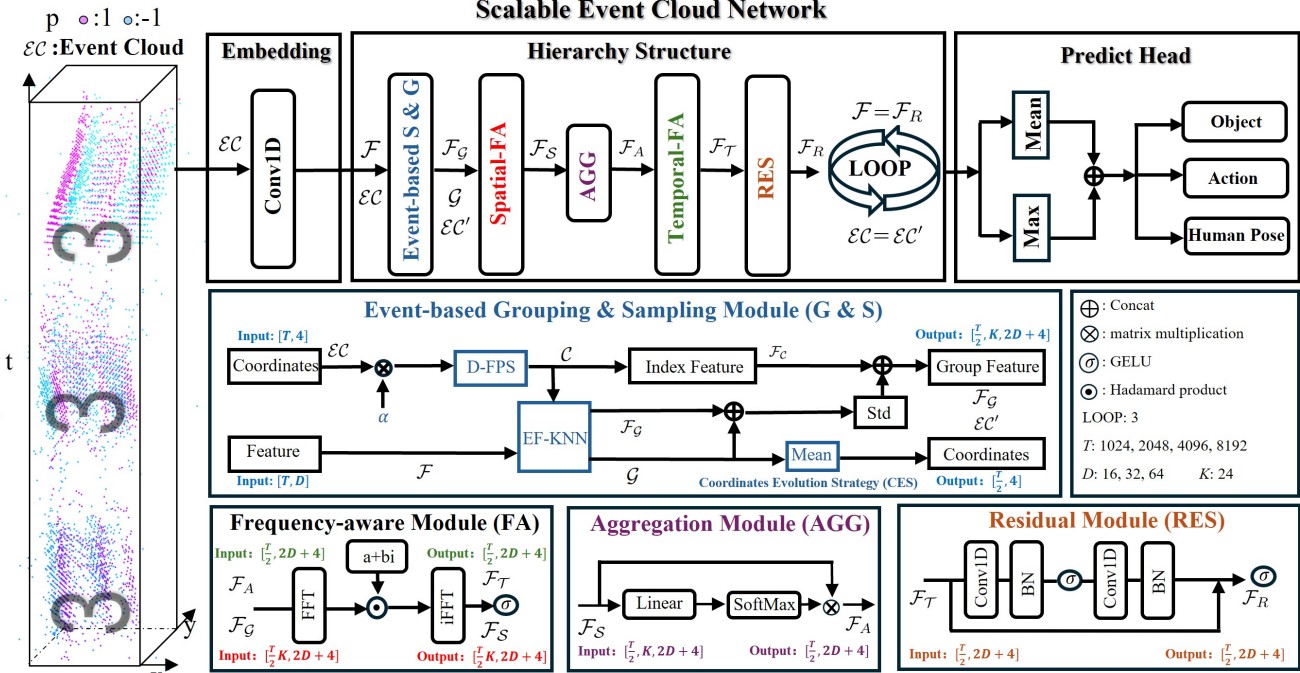

*Figure 3.* SECNet's architecture. It accomplishes three tasks by processing the Event Cloud into a sequence of distinct modules: Embedding, Hierarchy Structure, and Predict Head. In more detail, the Hierarchy Structure contains five different modules: Event-based G & S module captures the local neighborhood relationships, Spatial-FA abstracts the explicit spatial and implicit temporal features, AGG aggregates the features in a group, Temporal-FA catches the global explicit temporal features, and RES is responsible for the abstraction and high-level representation of features. The final results are obtained by the max-pooling and average-pooling features.

either counting the events or summing their polarity over time intervals (Gehrig et al., 2019; Cannici et al., 2020; Lee et al., 2016). However, this representation often results in a loss of temporal information and sparsity since it compresses dynamic information into static images (Wang et al., 2019a). To address this, Time Surface representations are proposed, which encode the timestamp of the most recent event at each pixel location, thereby maintaining some temporal dynamics (Sironi et al., 2018). Additionally, Bina-Rep converts asynchronous events to a sequence of sparse and expressive event frames (Barchid et al., 2022), but it is very time-consuming. Another advanced method is voxel-based representation, which segments the event data into multiple time bins, further capturing temporal features across 3D voxel grids or graphs that are beneficial for tasks like 3D reconstruction or depth estimation (Xie et al., 2022; Bi et al., 2020; Deng et al., 2022). Additionally, Point Cloud representation takes a different approach by representing the event data in a 3D spatial format, where each event's spatial and temporal attributes are preserved (Wang et al., 2019a; Ren et al., 2023). A more detailed comparison can be found at Section B. However, this representation ignores polarity and generally has a limited point number. Consequently, we embrace Event Cloud representation closest to the raw output, which only needs random downsampling.

## 2.2. Frequency-aware Learning

The Fourier transform is a powerful processing tool in computer vision. With the rapid advancement of convolutional neural networks, the theoretically substitutable frequency-domain Hadamard product offers superior global capture capability and computational efficiency. Initially, FFC (Chi et al., 2020) employed local Fourier units instead of traditional convolutions, performing convolution operations in the frequency domain to process image features at different scales effectively. Subsequently, GFNet (Rao et al., 2021) introduced a new technique that enhances feature expression by conducting element-wise multiplication between frequency domain features and learnable global filters. SpectFormer (Patro et al., 2023) enhanced the feature representation capabilities of the original ViT (Dosovitskiy, 2020) architecture by combining spectral analysis with multi-head self-attention mechanisms. Recently, network architectures based on frequency domain modules have been extensively applied to various sub-tasks, such as super-resolution (Li et al., 2018; Chen et al., 2024; Xue et al., 2020), low-level tasks (Mao et al., 2023; 2024; Yu et al., 2022; Kim et al., 2024), and human pose estimation (Zhao et al., 2023). Event Cloud is a collection of long-term signals containing rich spatial and temporal information,

and the utilization of frequency-domain feature extraction can better capture the global temporal relationship between events and conduct lightweight networks.

# 3. Method

## 3.1. Event Cloud Representation

Event cameras generate 2-Spatial, 1-Temporal, and 1-Polarity (2S-1T-1P) raw events in response to illumination changes in the environment, which can be defined:

$$\mathcal{E} = \{e_i = (x_i, y_i, t_i, p_i) \mid i \in I\}, I = [1, 2 \ldots, n], \quad (1)$$

where $(x, y)$ represents the spatial coordinate where the event emits, $t$ denotes the timestamps, $p$ means the polarity, and $i$ is the index representing $i$-th element in the event stream. As described in Section 2.1, $\mathcal{E}$ can be transformed into many representations. Still, Event Cloud is the closest representation to raw output and only requires random downsampling. Event Cloud can be generated by:

$$\mathcal{EC} = \{e_j = (x_j, y_j, t_j, p_j) \mid j \in J\}, J \subseteq_{\text{ord}} I, \quad (2)$$

where $J$ is a sequential subset of $I$, and the length of $J$ is the number of downsampling points. We define the length of $J$ as $T_0$, so the dimension of $\mathcal{EC}$ is $[T_0, 4]$. Specific settings for each dataset are shown in Table 1. It is crucial to emphasize that Event Cloud does not exhibit the permutation invariance characteristic of Point Cloud. So, the sampled events are arranged in the chronological order of their occurrence, enabling the network to extract explicit events' temporal features subsequently. Finally, Event Cloud is normalized by Min-Max and sent into the network.

## 3.2. Overall Architecture

To fully utilize the spatio-temporal richness and polarity of Event Cloud representation, we design a modular and hierarchical architecture. The overall framework consists of multiple iterative stages.

These operations are applied sequentially to gradually abstract features from sparse, asynchronous events while maintaining computational efficiency. The overall computational flow of the SECNet pipeline is formulated as follows:

$$\mathcal{G}_i, \mathcal{F}_{\mathcal{G}_i}, \mathcal{EC}' = \text{G\&S}(\mathcal{EC}, \mathcal{F}_{i-1}), \quad (3)$$

$$\mathcal{F}_{\mathcal{S}_i} = \text{SFA}(\mathcal{F}_{\mathcal{G}_i}), \quad \mathcal{F}_{A_i} = \text{AGG}(\mathcal{F}_{\mathcal{S}_i}), \quad (4)$$

$$\mathcal{F}_{\mathcal{T}_i} = \text{TFA}(\mathcal{F}_{A_i}), \quad \mathcal{F}_{R_i} = \text{RES}(\mathcal{F}_{\mathcal{T}_i}), \quad (5)$$

$$\mathcal{F}_i = \mathcal{F}_{R_i}, \mathcal{EC} = \mathcal{EC}', \text{where } i \in [1, m], \quad (6)$$

$$\mathcal{F}_{\text{end}} = \mathcal{F}_{R_m}, \mathcal{F}_0 = \text{Embed}(\mathcal{EC}). \quad (7)$$

Here, $\mathcal{EC}$ denotes the input Event Cloud, and $\mathcal{F}_i$ represents the feature set at the $i$-th stage, where $\mathcal{F}_0$ is obtained from embedding the initial Event Cloud. G&S$(\cdot)$ is the

Event-based Grouping and Sampling module that outputs the grouped coordinates $\mathcal{G}_i$, the grouped features $\mathcal{F}_{\mathcal{G}_i}$, and the updated Event Cloud $\mathcal{EC}'$. The Spatial Frequency-aware module is denoted by SFA$(\cdot)$, yielding spatially enhanced features $\mathcal{F}_{\mathcal{S}_i}$. The attention-based aggregation AGG$(\cdot)$ then produces $\mathcal{F}_{A_i}$. TFA$(\cdot)$ represents the Temporal Frequency-aware module, which captures long-range dependencies, producing $\mathcal{F}_{\mathcal{T}_i}$. This is refined through a residual module RES$(\cdot)$ to obtain the final output $\mathcal{F}_{R_i}$ for stage $i$. At the end of each stage $i$, the feature is updated via $\mathcal{F}_i = \mathcal{F}_{R_i}$, and the Event Cloud is replaced by $\mathcal{EC}'$ to reflect the centroid evolution. After $m$ stages, the final feature $\mathcal{F}_{\text{end}}$ is defined as $\mathcal{F}_{R_m}$, which is used for the subsequent classification and regression heads. Next, we elaborate on these innovative modules in the following subsections.

### 3.2.1. EVENT-BASED GROUPING & SAMPLING

Considering the inherent difference in the spatial $(x, y)$, temporal $t$, and polarity $p$ four coordinates within Event Cloud, we redesign the Grouping & Sampling (G & S) Module to faithfully extract those by treating them differently and incorporating polarity structurally. Point-based G & S module serves as the fundamental operation that captures local features with the limited receptive field in the Point Cloud space, akin to the convolutional operation in frame-based architecture. Most previous work has ignored the polarity and considered $t$ as $z$ of Point Cloud, where temporal information has been treated as an implicit spatial variable for feature extraction. However, the information represented by 2S-1T-1P is not wholly equivalent to that of Point Cloud. To address this discrepancy, we propose the Event-based G & S Module, the details of which will be discussed below.

Before being fed into the Event-based G & S module, Event Cloud passes through the embedding layer, as shown in the Figure 3. The coordinates information represented by $\mathcal{EC}$ with dimension $[T_0, 4]$ becomes a high-dimensional feature, defined as $\mathcal{F}_0$, with dimension $[T_0, D_0]$. Consequently, the inputs to the Event-based G & S Module are both $\mathcal{EC}$ and $\mathcal{F}_i$. Then, we downsample half events to find the centroid using Farthest Point Sampling (FPS) with the coordinates multiplied by a learnable scaling factor $\alpha$ to focus on different dimensions. We call this process Differentiation FPS (D-FPS), and this process is formulated by:

$$\mathcal{C}_i = \text{FPS}(\alpha \cdot \mathcal{EC}), \alpha \in \mathbb{R}^4, \mathcal{C}_i \in \mathbb{R}^{\frac{T_{i-1}}{2} \times 4}, \quad (8)$$

where $\mathcal{C}_i$ denotes the centroids' coordinates in $i$-th stage, which are utilized to group events. Then, the centroid $\mathcal{C}_i$ and features $\mathcal{F}_i$ are fed into the K-Nearest Neighbor (KNN) block (Wang et al., 2019b). Different from previous work utilizing coordinates to find the neighbors, we adopt the distance of events' features and call this process Event Features-based KNN (EF-KNN), which is formulated by

the following:

$$\mathcal{G}_i, \mathcal{F}_{\mathcal{G}_i} = \mathrm{KNN}(\mathcal{F}_{\mathcal{C}_i}, \mathcal{F}_i, K), \qquad (9)$$

where $K$ is the number of events in the group, $\mathcal{F}_{\mathcal{C}_i}$ represents the centroids' feature, $\mathcal{G}_i \in \mathbb{R}^{\frac{T_{i-1}}{2} \times K \times 4}$ denotes the grouped events' coordinate, and $\mathcal{F}_{\mathcal{G}_i} \in \mathbb{R}^{\frac{T_{i-1}}{2} \times K \times D_{i-1}}$ means the grouped events' features which is got by index feature from $\mathcal{F}_i$. The new coordinate is generated by the mean operation within the group, and we call this coordinate update process the Coordinates Evolution Strategy (CES),

$$\mathcal{EC}' = \mathrm{Mean}(\mathcal{G}_i), \mathcal{EC}' \in \mathbb{R}^{\frac{T_{i-1}}{2} \times 4}. \qquad (10)$$

The new group feature is concatenated by two elements:

$$\mathcal{F}_{\mathcal{G}_i} = \mathcal{F}_{\mathcal{G}_i} \oplus \mathcal{G}_i, \mathcal{F}_{\mathcal{G}_i} \in \mathbb{R}^{\frac{T_{i-1}}{2} \times K \times (D_{i-1}+4)}, \qquad (11)$$

where $\oplus$ means the concatenation operation. Finally, the new group feature $\mathcal{F}_{\mathcal{G}_i}$ is obtained under standardized and concatenated with the centroids' features. The standardized process utilizes the group mean feature as the mean value:

$$\mathcal{M} = \mathrm{Mean}(\mathcal{F}_{\mathcal{G}_i}), \mathcal{F}_{\mathcal{G}_i} = \frac{\mathcal{F}_{\mathcal{G}_i} - \mathcal{M}}{\mathrm{Std}(\mathcal{F}_{\mathcal{G}_i}, \mathcal{M})},$$
$$\mathcal{F}_{\mathcal{G}_i} = \mathcal{F}_{\mathcal{G}_i} \oplus \mathcal{F}_{\mathcal{C}_i}, \mathcal{F}_{\mathcal{G}_i} \in \mathbb{R}^{\frac{T_{i-1}}{2} \times K \times (2D_{i-1}+4)}, \qquad (12)$$

where $\mathcal{M} \in \mathbb{R}^{\frac{T_{i-1}}{2} \times (D_{i-1}+4)}$ denotes the mean values, and Std is the process to compute the standard deviation. With the deepening of the network, coordinates are decreasing and features are increasing, as shown in the Figure 4.

### 3.2.2. FREQUENCY-AWARE MODULE

We embrace frequency domain analysis to abstract the spatial and temporal features through Spatial Frequency-aware (FA) modules and Temporal-FA modules, respectively. Event Cloud contains both spatial and temporal information, and the performance is positively correlated to the number of events. Theoretically, more events contain finer spatial and temporal information. Nevertheless, **capturing the contextual relationships among a large number of events from space and time poses a challenge for existing event-based network models**, as demonstrated in Table 9. Simultaneously, the network has to be scalable and efficient to match the low power consumption of the event cameras for a wider range of applications. To address the performance plateau and the energy consumption issue with a large number of events, we adopt the Fourier transform for frequency domain feature extraction, with the process formulated as follows. The definition of spatio-temporal feature can be found Section K.

Different from images, Event Cloud is a one-dimensional time-series signal $[T_0, 4]$, as shown in Equation (2). One-dimensional discrete Fourier Transform (DFT) is employed

to convert the Event Cloud features to the frequency domain by the following formulation:

$$X[k] = \sum_{n=0}^{T-1} x[n] e^{-j\frac{2\pi}{T}kn}, \quad k = 0, 1, \ldots, T-1, \quad (13)$$

where $j$ is the imaginary unit, $x$ represents the different features ($\mathcal{F}_{\mathcal{G}_i}$ or $\mathcal{F}_{A_i}$) in the hierarchy structure, $X$ denotes the spectrum at different frequencies, and $T$ is the length of temporal signals $x$. Specifically, $\mathcal{F}_{\mathcal{G}_i}$ is reshaped to $[B \times T_i, D_i, K]$, where $B$ is batch size, while $\mathcal{F}_{A_i}$ retains its original shape of $[B, T_i, D_i]$, and the DFT is applied to the $D_i$ spatial dimension and $T_i$ temporal dimensions. Additionally, the inverse DFT can recover the spectrum from temporal signals by the following formulation:

$$x[n] = \frac{1}{T} \sum_{k=0}^{T-1} X[k] e^{j\frac{2\pi}{T}kn}, \quad n = 0, 1, \ldots, T-1, \quad (14)$$

Mathematically, the $X[T - k] = X^*[k]$, where $k \in [0, \frac{T}{2}]$, it means the spectrum $X$ is conjugate symmetric. Therefore, the transformed frequency domain spectrum only needs $\frac{T}{2} + 1$ long enough to be recovered to the original signal.

Once the spectrum is obtained through DFT, we can initialize a learnable filter $V$ with dimensionality matching that of the spectrum and perform the Hadamard product between the spectrum and the filter. In summary, the specific process of the frequency-aware module is as follows:

$$X = \mathrm{FFT}(x), \hat{X} = V \odot X, x = \mathrm{iFFT}(\hat{X}), x = \sigma(x), \qquad (15)$$

where $\sigma$ is a nonlinear activation function and $\odot$ means Hadamard product. As shown in Figure 3, SECNet adopts two different frequency modules, the Spatial-FA module and the Temporal-FA module. They perform Fourier transforms on Event Cloud features in the spatial $D_i$ and temporal $T_i$ dimensions, respectively, and work on extracting spatial and temporal features. The functions of these two modules will be discussed in detail in the following.

**Spatial-FA Module**. The input feature $\mathcal{F}_{\mathcal{G}_i}$ to this module is the output of the Event-based G & S module, with the dimension $[T_i, K, D_i]$ as shown in Equation (12). It represents the topological correlations composed of the spatial proximity, temporal continuity, and polarity relevance of K neighboring points within the group. The low frequencies correspond to globally consistent spatiotemporal patterns within groups, and high frequencies capture local details within neighborhood features. Previous approaches all employed MLP-based blocks to abstract features across the feature dimensions $D_i$, and this also means doing $T_i \times K$ repeating calculations. The computational complexity of a single MLP is $\mathcal{O}(D_i^2)$. We utilize frequency domain multiplication instead of MLP, and the complexity becomes

| Dataset | Source | Object | Resolution | Train | Test | Point Number | Class | Avg.length (ms) | SW (ms) |
|---|---|---|---|---|---|---|---|---|---|
| NMNIST (Orchard et al., 2015) | Synthetic | Static | 32x32 | 60000 | 10000 | 4096 | 10 | 300 | 300 |
| N-Caltech101 (Orchard et al., 2015) | Synthetic | Static | 240x180 | 20896 | 5226 | 8192 | 101 | 300 | 30 |
| CIFAR10-DVS (Li et al., 2017) | Synthetic | Static | 128x128 | 99965 | 24950 | 10240 | 10 | 1280 | 100 |
| N-Cars (Sironi et al., 2018) | Real | Static | 128x128 | 15422 | 8607 | 8192 | 2 | 100 | 100 |
| ASL-DVS (Bi et al., 2020) | Real | Static | 240x180 | 80640 | 20160 | 4096 | 24 | 100 | 100 |
| DVS128 Gesture (Amir et al., 2017) | Real | Dynamic | 128x128 | 26796 | 6959 | 1024 | 11 | 6520 | 500 |
| Daily DVS (Liu et al., 2021a) | Real | Dynamic | 346x260 | 2924 | 731 | 8192 | 12 | 3000 | 1500 |
| UCF101-DVS (Bi et al., 2020) | Synthetic | Dynamic | 240x180 | 108065 | 27384 | 8192 | 101 | 6600 | 1000 |
| THU$^{\text{E-ACT}}$-50 (Gao et al., 2023) | Real | Dynamic | 1280x800 | 62336 | 15107 | 8192 | 50 | - | 500 |
| DHP19 (Calabrese et al., 2019) | Real | Dynamic | 346x240 | 124980 | 24588 | 4096 | 13 | 128 | - |

*Table 1.* Specific details of different datasets. Point Number denotes the fixed downsampled value, and SW represents the sliding window.

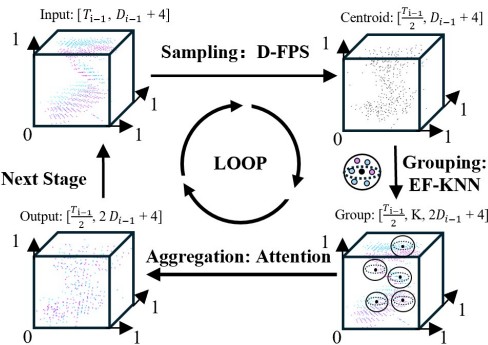

*Figure 4.* Visualization of Event Cloud's coordinates and features dimension during hierarchy structure. The closer to the end of the loop, the fewer the number of events and the greater the feature dimension.

$\mathcal{O}(D_i log_2(D_i))$. Taking the dimension 540 of the last stage in SECNet, the individual computational complexity is reduced by a factor of 60. This reduced complexity is finally multiplied by the number of repeats $T_i \times K \approx 12 \times 10^3$. The computational complexity reduction by using the Spatial-FA module is up to 20 times in the actually constructed network. More importantly, the network's performance on the various datasets does not give too much of a discount.

**Temporal-FA Module**. SECNet adopts a much larger number of input events $T_0$, which is several times as many as in previous Point Cloud-based works. Long-term events have important global and local contextual relationships. Commonly used models like LSTM and attention are not adept at capturing these relationships and would introduce excessive computational complexity, as demonstrated in Table 9. The input of the Temporal-FA Module is the output of the aggregation module. The aggregation (AGG) module employs a soft attention mechanism on the neighbor dimension to aggregate the feature within a group on the $t$ dimension.

$$A_i = \text{SoftMax}(\text{MLP}(\mathcal{F}_{S_i})), A_i \in \mathbb{R}^{T_i \times K}, \quad (16)$$

$$\mathcal{F}_{A_i} = A_i \cdot \mathcal{F}_{S_i}, \mathcal{F}_{A_i} \in \mathbb{R}^{T_i \times D_i}, \quad (17)$$

where $A_i$ is the attention value of group members, and $\mathcal{F}_{S_i}$ is the output of Spatial-FA module. Then, SECNet

performs a Fourier transform in the time dimension $T_i$ of the event features $\mathcal{F}_{A_i}$. And the temporal features of different frequencies are captured through Equation (15). The filters $V$ are global since they can cover all the frequencies and capture both long-term and short-term interactions.

In summary, SFA is applied to the grouped feature after the Event-based G & S, where each group represents a local neighborhood formed by spatial proximity, temporal continuity, and polarity relevance. It can efficiently model intragroup local structure, serving as a lightweight replacement for repeated MLP-based local feature abstraction. In contrast, TFA is introduced after the aggregation stage, when the neighborhood dimension has already been collapsed, and the events remain chronologically ordered. Its role is to model inter-event temporal dependencies over long event streams, rather than local neighborhood interactions.

### 3.2.3. RESIDUAL MODULE

To abstract the higher-level features within the network from Event Cloud, we employ a basic residual module at the end to further enhance the complexity and hierarchy of features, as shown in Figure 3. The pseudo code can be found in Section G and specific dimension variance shown in Section F.

## 4. Experiment

### 4.1. Implementation Details

The hardware platform configuration comprises the following: CPU: AMD 7950x, GPU: RTX 4090. Our training model employs the subsequent set of fixed hyperparameters. Optimizer (AdamW), Initial Learning Rate (0.001), Scheduler (Cosine), and Maximum Epochs (150). The Batch Size is dynamically set to take full use of the GPU's memory.

The **Accuracy** metric used in the object classification and action recognition tasks represents the ratio of correctly predicted samples to the total samples. The **Mean Per Joint Position Error** (MPJPE) metric is used in the human pose estimation tasks to evaluate the average Euclidean distance between the ground truth and prediction, and the metric

| Method | Type | Publish | N-MNIST | N-CARS | CIFAR10-DVS | N-Caltech101 | ALS-DVS | Average |
|---|---|---|---|---|---|---|---|---|
| | | | Pretrained on ImageNet | | | | | |
| EST (Gehrig et al., 2019) | Frame | ICCV'19 | 0.991 | 0.925 | 0.749 | 0.837 | 0.991 | 0.899 |
| M-LSTM (Cannici et al., 2020) | Frame | ECCV'20 | 0.989 | 0.957 | 0.73 | 0.857 | 0.992 | 0.905 |
| MVF-Net (Deng et al., 2021) | Frame | TCSVT'21 | 0.993 | 0.968 | 0.762 | 0.871 | 0.996 | 0.918 |
| | | | Without Pretrainning | | | | | |
| EST (Gehrig et al., 2019) | Frame | ICCV'19 | 0.99 | 0.919 | 0.634 | 0.753 | 0.979 | 0.855 |
| M-LSTM (Cannici et al., 2020) | Frame | ECCV'20 | 0.986 | 0.927 | 0.631 | 0.738 | 0.98 | 0.852 |
| MVF-Net (Deng et al., 2021) | Frame | TCSVT'21 | 0.981 | 0.927 | 0.599 | 0.687 | 0.971 | 0.833 |
| AsyNet (Messikommer et al., 2020) | Frame | ECCV'20 | - | 0.944 | 0.663 | 0.745 | - | - |
| Gabor-SNN (Lee et al., 2016) | Frame | CVPR'18 | 0.837 | 0.789 | 0.245 | 0.196 | - | - |
| HATS (Sironi et al., 2018) | Frame | CVPR'18 | 0.991 | 0.902 | 0.524 | 0.642 | - | - |
| ECSNet (Chen et al., 2022b) | Frame | TCSVT'22 | 0.992 | 0.946 | 0.727 | 0.693 | 0.997 | 0.871 |
| RG-CNNs (Bi et al., 2020) | Voxel Graph | TIP'20 | 0.99 | 0.914 | 0.54 | 0.657 | 0.901 | 0.800 |
| EV-VGCNN (Deng et al., 2022) | Voxel Graph | CVPR'22 | 0.994 | 0.953 | 0.651 | 0.748 | 0.983 | 0.866 |
| EVSTr (Xie et al., 2024) | Voxel Grid | TCSVT'24 | - | 0.941 | 0.731 | 0.797 | 0.997 | - |
| VMV-GCN (Xie et al., 2022) | Voxel Graph | RAL'22 | 0.995 | 0.932 | 0.69 | 0.778 | 0.989 | 0.877 |
| EDGCN (Deng et al., 2024) | Voxel Graph | AAAI'24 | - | **0.958** | 0.716 | 0.801 | - | - |
| VMST-Net (Liu et al., 2023) | Voxel Grid | TCSVT'24 | 0.995 | 0.944 | 0.753 | 0.822 | 0.998 | 0.902 |
| EventNet (Sekikawa et al., 2019) | Point Cloud | CVPR'19 | 0.752 | 0.75 | 0.171 | 0.425 | 0.833 | 0.586 |
| PointNet++ (Qi et al., 2017b) | Point Cloud | NIPS'17 | 0.841 | 0.809 | 0.465 | 0.503 | 0.947 | 0.713 |
| SECNet | Event Cloud | Our | **0.997** | 0.947 | **0.757** | **0.824** | **0.9993** | **0.905** |

*Table 2.* Experiment on event-based object classification datasets.

| Method | Type | DVSGesture | DailyDVS | UCF101-DVS | Average |
|---|---|---|---|---|---|
| TANet (Liu et al., 2021b) | F | 0.974 | 0.963 | 0.669 | 0.869 |
| I3D (Carreira & Zisserman, 2017) | F | 0.951 | 0.909 | 0.603 | 0.822 |
| ECSNet (Chen et al., 2022b) | F | 0.986 | - | 0.702 | - |
| TimeSformer (Bertasius et al., 2021) | F | 0.917 | 0.906 | 0.541 | 0.788 |
| RG-CNN (Bi et al., 2020) | V | 0.968 | - | 0.678 | - |
| VMV-GCN (Xie et al., 2022) | V | 0.975 | 0.941 | - | - |
| EVSTr (Xie et al., 2024) | V | 0.986 | 0.996 | 0.735 | 0.906 |
| VMST-Net (Liu et al., 2023) | V | 0.978 | - | 0.787 | - |
| TTPOINT (Ren et al., 2023) | PC | 0.988 | 0.991 | 0.725 | 0.901 |
| SpikePoint (Ren et al., 2024) | PC | 0.988 | 0.979 | 0.685 | 0.884 |
| EventMamba (Ren et al., 2025) | PC | **0.992** | 0.991 | 0.903 | 0.962 |
| SECNet | EC | 0.989 | **0.9965** | **0.916** | **0.967** |

*Table 3.* Event-based action recognition dataset Results.

| Method | N-Caltech101 | Daily DVS | DHP19 |
|---|---|---|---|
| Event Frame | 4.2ms | 2.2ms | 0.5ms |
| Binary Map | 772ms | 286ms | 717ms |
| Time Surface | 4.3ms | 3.1ms | 3.3ms |
| Voxel Grid | 14.6ms | 7.9ms | 4.3ms |
| Rasterized Point Cloud | 5.8ms | 4ms | 2.9ms |
| Event Cloud | **1.5ms** | **1ms** | **0.3ms** |

*Table 4.* Preprocessing time of different representations.

space for 2D error is pixels and millimeters for 3D error. The number of skeleton key points set for the experiment is 13. More details can be found in Sections A, C and D.

### 4.2. Preprocessing time

We utilize tonic (Lenz et al., 2021) as a code framework, which is a powerful tool for event datasets and transforms. As shown in Table 4, the time of preprocessing from raw event output to six representations is required on three datasets. It's intuitive to see that Event Cloud takes the

least amount of time, and this time barely affects the overall throughput of the model. The Voxel representations, also equipped with lightweight models, spend a lot of time in data format transformation, nearly 10 times more than Event Cloud. This time far exceeds that of network inference, which hinders the practical deployment of the algorithm.

### 4.3. Object Classification Results

Object classification involves making irregular movements with an event camera around the recognized object, with the classification information primarily encoded in $(x, y)$ coordinates of the event. Theoretically, stacking events into frames would make it easier to reach the performance plateau. But without the pre-training process of ImageNet, the frame-based method performs inferior to the point-based method, which contains the Voxel, Point Cloud, and Event Cloud as shown in Table 2. This is due to the fact that mapping sparse data onto an intensity 2D plane may have aberrations such as ghosting and blurring, which reduce

| Method | HMAX SNN | Motion SNN | EV-ACT | SECNet |
|---|---|---|---|---|
| Accuracy | 65.1 | 85.3 | 92.7 | **97.25** |

*Table 5.* Experiment on THU$^{\text{E-ACT}}$-50 dataset.

| Name | Type | Param. | GMACs | FPS | Acc. |
|---|---|---|---|---|---|
| C3D | F | 78.41 | 19.85 | 18 | 0.472 |
| ResNext-50 | F | 26.05 | 3.23 | 35 | 0.602 |
| 3D-ResNet | F | 63.5 | 14.99 | - | 0.579 |
| I3D | F | 12.37 | 15.06 | 34 | 0.635 |
| EventLSTM | F | 21.4 | - | - | 0.776 |
| ECSNet | F | - | 6.12 | 0.36 | 0.702 |
| RG-CNN | V | 6.95 | 12.46 | - | 0.632 |
| EVSTr | V | 2.88 | 1.38 | 41 | 0.735 |
| VMST-Net | V | 3.61 | 0.31 | 23 | 0.787 |
| TTPOINT | P | 0.357 | 0.294 | 88 | 0.725 |
| EventMamba | P | 3.28 | 1.861 | 69 | 0.903 |
| SECNet | EC | 1.126 | 0.48 | 120 | 0.916 |

*Table 6.* Hardware benchmark on UCF101-DVS dataset. We adopt the dataset filename-based partitioning method (Ren et al., 2025).

| Method | Type | MPJPE$_{2D}$ | MPJPE$_{3D}$ | Param. | GMACs |
|---|---|---|---|---|---|
| Pose-Res50 | F | 5.28 | 59.83 | 34 | 12.91 |
| LeVit-128S | F | 7.68 | 87.79 | 7.87 | 0.2 |
| DHP19 | F | 7.67 | 87.9 | 0.22 | 3.51 |
| VMV-PointTrans | V | 9.13 | 103.23 | 3.67 | 5.98 |
| VMST-Net | V | 6.45 | 73.07 | 3.59 | 0.38 |
| PointNet | RPC | 7.29 | 82.46 | 4.46 | 1.19 |
| DGCNN | RPC | 6.83 | 77.32 | 4.51 | 4.91 |
| Point Transformer | RPC | 6.46 | 73.07 | 3.65 | 5.03 |
| SECNet | EC | **6.11** | **69.89** | **1.831** | **0.160** |

*Table 7.* Experiment on event-based human pose dataset. RPC means the rasterized Point Cloud representation (Zhao et al., 2021).

| Dataset | DHP19 | | N-CARS | UCF101-DVS |
|---|---|---|---|---|
| Metric | MPJPE$_{2D}$ | MPJPE$_{3D}$ | Acc. | Acc. |
| w all | **6.11** | **69.89** | **0.947** | **0.916** |
| w/o $\alpha$ | 6.17 | 70.03 | 0.942 | 0.914 |
| w/o EF-KNN | 6.20 | 70.47 | 0.937 | 0.915 |
| w/o SFA | 6.68 | 75.43 | 0.929 | 0.892 |
| w/o TFA | 6.47 | 73.84 | 0.944 | 0.913 |
| Input P | 6.27 | 71.12 | 0.937 | 0.913 |
| w/o P | 6.19 | 70.30 | 0.944 | 0.911 |

*Table 8.* Ablation study on core modules.

the quality of the dataset. Prior to our work, Voxel representations dominated such tasks. Point Cloud has limited scalability to represent objects, resulting in consistently poor performance, which EventNet and PointNet++ demonstrate. SECNet adopted Event Cloud representation to improve performance by 27% compared to the state-of-the-art (SOTA) Point Cloud method. Even compared with the Voxel-based SOTA methods, SECNet shows comparable results on all object classification datasets.

### 4.4. Action Recognition Results

Action recognition is a task used to analyze human movements or gestures. It is widely applied in various fields, including sports, security, and entertainment. This task differs from object classification in that the main information is embedded in the $t$ coordinates, which represent the type of motion of the observed person or hand. However, voxel-based and frame-based representations tend to diminish the resolution of $t$, reducing it from a fine-grained range (1 $\mu s$) to a coarse-grained range (50 $ms$). So, the finer-grained temporal resolution of the representations, the better the performance results, as shown in Table 3. SECNet utilizes the finest-grained representation of Event Cloud to fully extract its spatial-temporal features, achieving SOTA results on average across three action recognition datasets. Especially on the UCF101-DVS dataset, SECNet far outperforms the previous SOTA methods by 16%, named VMST-Net. **This implies that critical information you omit in the representations, and you cannot recover or make it up through the network architecture**. Additionally, we show the hardware performance in UCF101-DVS in Table 6.

**Scalability to High-resolution dataset**. SECNet outperforms all other methods by 5%, as presented in Table 5. Moreover, compared to EV-ACT (Gao et al., 2023),

SECNet consumes one order fewer hardware resources (1.27 M/ 0.971 MACs vs 21.3 M/ 14.5 MACs). It is also evident that the required computation resource is independent of the resolution, making it particularly suitable for processing high-resolution event camera data.

### 4.5. Human Pose Estimation Results

Human pose estimation is a fundamental task that is widely used in virtual reality and security. It aims to accurately regress the human skeletons' key points. Event cameras are distinguished by their advantages in privacy, high speed, and ability to capture motion without blur in such tasks. Point Cloud representation was demonstrated to be efficient for edge devices, but did not perform well in the MPJPE metric. Furthermore, this work proposed the Rasterized Point Cloud representation (RPC), which retains the original Point Cloud network while additionally introducing the representation transformation time, as shown in Table 4. SECNet leverages the raw unprocessed Event Cloud to reach the SOTA results of point-based and voxel-based methods. Meanwhile, SECNet has the smallest model size, with GMACs of only 0.160, which is 2% of Point Transformer, as shown in Table 7. In addition, it takes only 4.37 ms for the model to infer a sample on the server. Furthermore, the citations corresponding to Tables 5 to 7 are provided in the Section A and visualization results shown in Section K.

### 4.6. Ablation Study

#### 4.6.1. MODULES ABLATION

In Table 8, "w/o $\alpha$ means SECNet does not adopt D-FPS. And "w/o EF-KNN" means SECNet does not adopt EF-

| Event number | SECNet | SECNet w LSTM | PointNet++ | TTPOINT |
|---|---|---|---|---|
| 512 | 46.29 | 44.16 | 37.63 | 39.70 |
| 1024 | 50.36 | 48.31 | 43.54 | 42.15 |
| 2048 | 52.24 | 49.10 | 47.02 | 42.58 |
| 4096 | 56.57 | 50.84 | 49.04 | 44.39 |
| 8192 | 60.04 | 53.63 | 53.15 | 43.65 |
| 10240 | 60.60 | 52.42 | 54.39 | 45.61 |

*Table 9.* Ablation study of different Point-based networks in N-Caltech101 with a 300 ms sliding window.

KNN. The performance of the model decreased on all three datasets. "w/o SFA" and "w/o TFA" denote that SECNet is not equipped with the Spatial-FA module and Temporal-FA module, respectively. The model showed a significant decrease in the DHP19 dataset, suggesting that temporal features are important for predicting the human skeleton. Input P means SECNet incorporates polarity in the input-level instead of the structural-level."w/o P" represents the input equal to the Point Cloud representation without polarity. All three datasets have demonstrated that polarity provides additional information. What's more, the more ablation study can be found in the Sections I and J. Overall, all the designed components are indispensable and jointly contribute to the superior prediction capability of our proposed SECNet.

### 4.6.2. SCALABILITY TO EVENT NUMBER

PointNet++ and TTPOINT do not have an explicit temporal feature extraction module; the accuracy is much lower than that of SECNet, and the performance of TTPOINT does not improve clearly after 4096 points, as shown in Table 9. Additionally, SECNet replaces LSTM for temporal feature extraction, which makes it difficult to capture long-term dependencies between events as the number of points grows. In contrast, the proposed Temporal-FA module models global temporal relationships in the frequency domain, enabling SECNet to effectively capture long-range dependencies while maintaining stable scalability as the event number increases. Moreover, the consistent performance improvement under larger event inputs indicates that SECNet can better utilize the fine-grained temporal information contained in dense Event Cloud representations.

### 4.7. Hardware Efficiency and Edge Deployability

we perform measured FPGA experiments on a Zynq UltraScale+ MPSoC running at 200 MHz to evaluate edge-oriented deployment characteristics. We compare the proposed SFA module against the conventional spatial MLP implementation used for feature extraction.

As shown in Table 10, the spatial MLP requires 6,302,723 / 12,595,715 / 25,191,430 cycles for stages 1/2/3, corresponding to 31.51 / 62.98 / 125.96 ms latency, respectively. In contrast, the proposed FFT-based SFA accelerator requires only 393,859 cycles (1.97 ms), and the cost remains nearly constant across different stages. This corresponds to approx-

*Table 10.* FPGA comparison between spatial MLP and SFA on Zynq UltraScale+ MPSoC (200 MHz).

| Input Dimension | MLP (Cycle Count / Latency) | SFA (Cycle Count / Latency) |
|---|---|---|
| Stage 1 [1024,24,64] | 6,302,723 / 31.51 ms | 393,859 / 1.97 ms |
| Stage 2 [512,24,128] | 12,595,715 / 62.98 ms | 393,859 / 1.97 ms |
| Stage 3 [256,24,256] | 25,191,430 / 125.96 ms | 393,859 / 1.97 ms |
| On-Chip Memory (RAMB18) | 84 | 39 |

imately $16\times$, $32\times$, and $64\times$ fewer cycles compared with the spatial baseline. Additionally, the on-chip memory RAM18 is reduced from 84 s to 39. This demonstrate SECNet is not only algorithmic lightweight, but also hardware-efficient realizability.

## 5. Conclusion

In this paper, we carry forward the Event Cloud representation, which is the nearest one to the raw events. We propose event-based G&S modules and embrace frequency domain analysis to extract long sequence events' features. We benchmark SECNet on ten datasets, and the ground-breaking results demonstrate scalability, efficiency, and effectiveness.

## Acknowledgements

This work was supported in part by the Youth Science and Technology Talent Support Program of GDSTA (SKXRC2025460), the Guangdong Science and Technology Program (2025A0505000036), the Guangdong Basic and Applied Basic Research Foundation (2026A1515010184), the National Science Foundation of China [U23A20389, 62306095, 82441009, 82441008, 62506101], the Heilongjiang Natural Science Foundation of China [LH2024F017], the China Postdoctoral Science Foundation [2024M764190], and the Fundamental Research Funds for the Central Universities [HIT.NSFJG202439, HIT.NSFJG202434].

## Impact Statement

This paper aims to advance lightweight and scalable event-based framework for edge applications. By leveraging Event Cloud representation and efficient frequency-domain feature extraction, SECNet enables low-latency and low-power event processing, which is particularly beneficial for resource-constrained scenarios such as robotics, wearable devices, autonomous systems, and smart sensing platforms.

As with other event-based perception technologies, the proposed method could potentially be applied in surveillance or monitoring systems. However, SECNet itself is a general-purpose vision backbone and does not introduce additional privacy risks beyond existing event-camera technologies. We hope this work encourages further research toward efficient, reliable, and responsible edge AI systems.

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

*Table 11.* Different Model Architecture Comparison

| Aspect | SECNet | PointNet++(Qi et al., 2017b) | EventNet(Sekikawa et al., 2019) | VMV-GCN(Xie et al., 2022) | VMST-Net(Liu et al., 2023) |
|---|---|---|---|---|---|
| Scalability | 10240 | 1024 | - | 2048 | 1024 |
| Representation | Event Cloud | Point Cloud | Event Cloud | Voxel Graph + Muti-view | Voxel Grid |
| Preprocessing Time | short | short | short | long | long |
| Polarity | Structural-level Integration | ✗ | Input-level Inclusion | Input-level Inclusion | Input-level Inclusion |
| Sampling Strategy | D-FPS | FPS | ✗ | Motion-sensitive | Fixed-grid |
| KNN Distance | Feature-based | Radius-based | ✗ | Geometry-based | Radius-based |
| Coord. Update | ✓(CES) | ✗ | ✗ | ✗ | ✗ |
| Long Sequence Model | ✓ | ✗ | ✗ | ✗ | ✗ |

## A. References to comparative method abbreviations

In Table 5, which presents the experiment on the THU$^{E-ACT}$-50 dataset, we include HMAX SNN (Xiao et al., 2019), Motion SNN (Liu et al., 2021a), and EV-ACT (Gao et al., 2023).

In Table 6, the hardware benchmark on the UCF101-DVS dataset, lists C3D (Fan et al., 2016), ResNext-50 (Hara et al., 2018), 3D-ResNet (Hara et al., 2018), I3D (Carreira & Zisserman, 2017), EventLSTM (Annamalai et al., 2022), ECSNet (Chen et al., 2022b), RG-CNN (Bi et al., 2020), EVSTr (Xie et al., 2024), VMST-Net (Liu et al., 2023), TTPOINT (Ren et al., 2023), and EventMamba (Ren et al., 2025).

In Table 7, the experiment on the event-based human pose dataset. We include Pose-Res50 (Xiao et al., 2018), LeVit-128S (Graham et al., 2021), DHP19 (Calabrese et al., 2019), VMV-PointTrans (Xie et al., 2022; Zhao et al., 2021), VMST-Net (Liu et al., 2023), PointNet (Chen et al., 2022a; Qi et al., 2017a), DGCNN (Chen et al., 2022a; Wang & Solomon, 2021), and Point Transformer (Chen et al., 2022a; Zhao et al., 2021).

## B. Comparison of Event Cloud and Voxel Representations

While Voxel-based (voxel grid and voxel graph) representations can indeed encode a high density of events and preserve polarity through multi-channel binning, they inherently impose a grid-like quantization over the spatiotemporal domain. This discretization leads to two limitations: (1) loss of temporal precision due to uniform binning, and (2) inefficiencies in data transformation and memory usage, especially for high-resolution or long-duration events. In contrast, Event Cloud maintains a continuous representation of each event with no spatial or temporal quantization, preserving the raw structure and allowing direct modeling without costly pre-processing time. Voxel-based representation often requires longer preprocessing time and introduces additional time for constructing graph structures.

To further compare from the network architecture perspective, voxel-based models such as VMV-GCN (Xie et al., 2022) and VMST-Net (Liu et al., 2023) typically rely on static receptive fields or geometry-based neighborhood construction, which are limited in their ability to adapt to dynamic spatial-temporal patterns. Their grouping mechanisms are predefined by voxel grid boundaries or spatial distances, ignoring the learned feature similarity between tokens. Moreover, **polarity information, although present at the input level, is only utilized in voxel-wise feature encoding, without participating in structural decisions such as neighborhood formation or coordinate update**.

In contrast, SECNet introduces a feature-based G&S mechanism that leverages both spatial-temporal positions and polarity in a unified 4D feature space. The D-FPS and EF-KNN enable the network to construct adaptive neighborhoods, while CES allows the model to dynamically adjust token positions layer-by-layer based on learned representations. This architectural design not only enhances the semantic alignment of local regions but also improves scalability to large and dense event streams by avoiding fixed voxel grids. Furthermore, SECNet uniquely integrates polarity information into its structural pipeline, making it the only model among the compared methods that fully exploits polarity for both feature and topology modeling. Specific comparison can be seen in Table 11.

## C. Implement Details

For all datasets, the hierarchy structure is configured with 3 loops, and the embedding layer is implemented using a single layer of one-dimensional convolution, with input and output dimensions of 4 and 64, respectively. So the dimensional transformation of Event Cloud features is gradually increased to 64, 132, 268, and 540 as the loop goes on, and the formula is calculated as the previous layer's dimension multiplied by two plus four. As for the number of groups, which is also the number of centroids in the first loop, NMNIST, N-CARS, DVS128 Gesture, and DHP19 are set to 512, ASL-DVS, Daily

| Events Number\Centroid | [2048, 1024, 512] | | | [1024, 512, 256] | | | [512, 256, 128] | | | [256,128,64] | | |
|---|---|---|---|---|---|---|---|---|---|---|---|---|
| | Runtime | Param. | GMACs | Runtime | Param. | GMACs | Runtime | Param. | GMACs | Runtime | Param. | GMACs |
| 10240 | 19.71 | 1.514 | 1.198 | 9.70 | 1.103 | 0.548 | 5.08 | 0.898 | 0.262 | 4.43 | 0.795 | 0.129 |
| 8192 | 16.8 | 1.514 | 1.063 | 8.34 | 1.103 | 0.48 | 4.37 | 0.898 | 0.228 | 4.43 | 0.795 | 0.112 |
| 4096 | 10.8 | 1.514 | 0.794 | 5.30 | 1.103 | 0.345 | 4.43 | 0.898 | 0.160 | 4.40 | 0.795 | 0.077 |
| 2048 | - | - | - | 4.44 | 1.103 | 0.278 | 4.38 | 0.898 | 0.126 | 4.35 | 0.795 | 0.060 |
| 1024 | - | - | - | - | - | - | 4.35 | 0.898 | 0.109 | 4.35 | 0.795 | 0.051 |

*Table 12.* Hardware performance lookup table for SECNet.

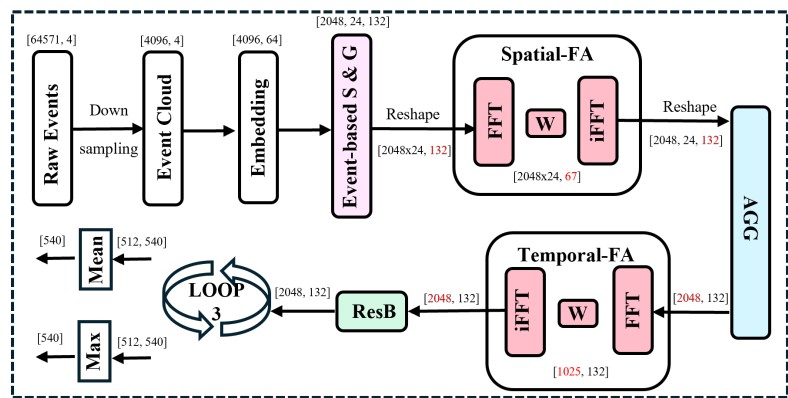

*Figure 5.* Specific variation of dimensions of a sample in SECNet.

DVS, and UCF101-DVS are set to 1024, and CIFAR10-DVS and N-Caltech101 are set to 2048. The number of groups is halved as each loop moves to the next one.

## D. Dataset

We validate SECNet on three tasks ten datasets, e.g., **Object Classification**: NMNIST (Orchard et al., 2015), N-Caltech101 (Orchard et al., 2015), CIFAR10-DVS (Li et al., 2017), N-CARS (Sironi et al., 2018), ASL-DVS (Bi et al., 2020), **Action Recognition**: DVS128 Gesture (Amir et al., 2017), Daily DVS (Liu et al., 2021a), UCF101-DVS (Bi et al., 2020), THU$^{\text{E-ACT}}$-50 (Gao et al., 2023), **Human Pose Estimation**: DHP19 (Calabrese et al., 2019). For N-Caltech101, CIFAR10-DVS, ASL-DVS, Daily DVS, and UCF101-DVS do not have official divisions, we follow the setting in (Bi et al., 2020; Chen et al., 2022b), randomly select 20% of the samples as the testset, and all other datasets use official divisions. The details of these datasets are shown in Table 1.

## E. Super-linear Increase

PointNet exhibits a linear increase in MACs with the number of points, as it only extracts global information, which is insufficient for complex tasks. In fact, mainstream models such as PointNet++, PointMLP, PointTransformer, and our SECNet extract local features and are also influenced by the number of FPS points (i.e., the number of groups), leading to superlinear growth in computational complexity as shown in Table 12. Still, SECNet decreases the MACs by 5x given the same sample points.

## F. Specific Dimension Variance in SECNet

We show the specific variation of the dimension of a sample in Figure 5. The red marks represent the dimensions done FFT. 1. Spatial-FA: This module functions on channel dimension (132), which represents spatial information between each event point and the central feature. Unlike the frame representation, the feature does not have x and y dimensions but rather abstracts the information of x and y into the channel dimension. 2. Temporal-FA: This module functions on the event number dimension (2048), which represents discretized temporal information due to its being strictly arranged in the order of events emitted.

---

**Algorithm 1** SECNet Pipeline

---

**Input**: Event Cloud $\mathcal{EC}$
**Parameters**: Stage Number: $m$
**Output**: Category $\hat{c}$, Human Pose $\hat{p}$

1: $\mathcal{F} = \text{Embedding}(\mathcal{EC})$,
2: **for** i in stage$(m)$ **do**
3:     **Event-based G & S Module**
4:     $\mathcal{G}, \mathcal{F}_{\mathcal{G}}, \mathcal{EC}' = \text{G\&S}(\mathcal{EC}, \mathcal{F})$
5:     **Spatial-FA Module**
6:     $\mathcal{F}_{\mathcal{S}} = \text{SFA}(\mathcal{F}_{\mathcal{G}})$;
7:     **Temporal Aggregate Module**
8:     $\mathcal{F}_A = \text{AGG}(\mathcal{F}_{\mathcal{S}})$;
9:     **Temporal-FA Module**
10:     $\mathcal{F}_{\mathcal{T}} = \text{TFA}(\mathcal{F}_A)$;
11:     **ResB Module**
12:     $\mathcal{F}_R = \text{RES}(\mathcal{F}_T)$
13:     $\mathcal{F} = \mathcal{F}_R$; $\mathcal{EC} = \mathcal{EC}'$;
14: **end for**
15: $\mathcal{F} = \text{Mean}(\mathcal{F}) \oplus \text{Max}(\mathcal{F})$
16: **Classifier**         **Regressor**
17: Get category $\hat{c}$     Get 13 skeleton points $\hat{p}$

---

## G. Hardware Resource Performance

In addition, we tested the hardware resource performance and runtime of SECNet on the server platform. Thanks to the design of the Spatial-FA module, SECNet achieves leading results with a remarkably low computational cost, requiring only one fortieth of the MACs compared to frame-based methods. Additionally, SECNet inherits the high inference speed of Point Cloud representations, delivering an impressive throughput of 229.7 FPS while far superior to frame-based and voxel-based methods. Although the number of parameters in SECNet is higher than that of TTPOINT, the tensor decomposition technique employed significantly reduces the network's inference time. SECNet still maintains a relatively low parameter count compared to the frame-based method.

## H. Pseudo Code

The SECNet pipeline is shown in Algorithm 1. The input is an Event Cloud along with the stage number parameter $m$. The output consists of the predicted category $\hat{c}$ and human pose $\hat{p}$.

First, the algorithm embeds the input Event Cloud using the embedding module, producing the feature representation $\mathcal{F}$. Then, the algorithm proceeds through a series of stages ($m$ iterations), where each stage consists of several modules. The Event-based G & S module processes the input $\mathcal{EC}$ and the embedded features $\mathcal{F}$ via the G & S function, generating group set $\mathcal{G}$, group features $\mathcal{F}_{\mathcal{G}}$, and a modified Event Cloud $\mathcal{EC}'$. Next, the Spatial Frequency-aware (SFA) module aggregates the feature graph $\mathcal{F}_{\mathcal{G}}$ spatially to produce the spatially aggregated features $\mathcal{F}_{\mathcal{S}}$. Afterward, the Temporal Aggregate module temporally aggregates the spatially aggregated features $\mathcal{F}_{\mathcal{S}}$ to produce the temporally aggregated features $\mathcal{F}_A$. Then, the Temporal Feature Aggregation (TFA) module processes the temporally aggregated features $\mathcal{F}_A$ to generate the temporal features $\mathcal{F}_{\mathcal{T}}$. Finally, the ResB module applies residual learning to the temporal features $\mathcal{F}_{\mathcal{T}}$ to obtain the final features $\mathcal{F}_{\mathcal{R}}$.

After each stage, the feature set $\mathcal{F}$ is updated with the residual features $\mathcal{F}_{\mathcal{R}}$, and the Event Cloud $\mathcal{EC}$ is updated with the modified Event Cloud $\mathcal{EC}'$. Upon completion of all stages, the algorithm computes the mean and maximum of the feature set $\mathcal{F}$ and combines them using a concat operation $\oplus$. Finally, a classifier regressor is applied to the fused features to predict the category $\hat{c}$ and extract 13 skeleton points $\hat{p}$ corresponding to the human pose.

This pipeline involves a series of feature transformations and aggregations aimed at efficiently and effectively processing event-driven data for both category classification and human pose estimation.

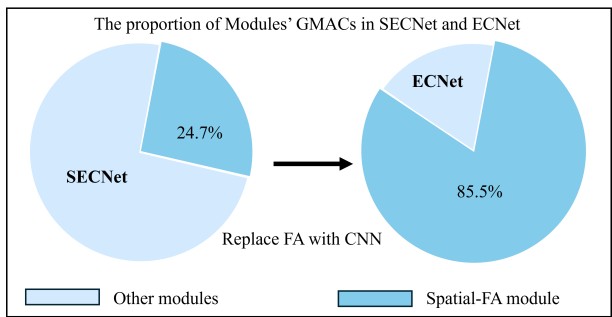

*Figure 6.* Proportional chart of GMACs. The other modules' GMACs, represented by the light blue, are congruent.

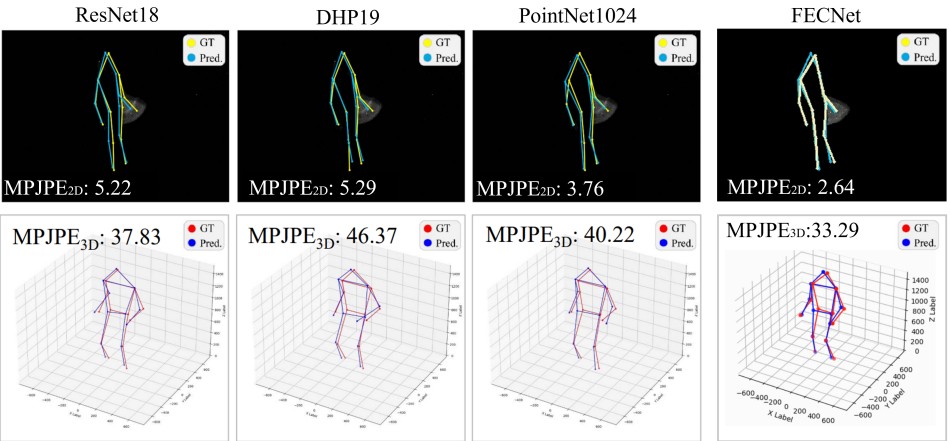

*Figure 7.* Human pose estimation results in visualization for different models. The first row shows the results in 2D (yellow for ground truth, blue for prediction). The second row shows the results in 3D (red for ground truth, blue for prediction).

## I. Ablation Study on Spatial-FA Module

As mentioned in the main paper, we replace the convolution operations with Spatial-FA, and this significantly reduces the overall GMACs of the model. To explore its effect on the results, we utilize convolution to extract spatial features on six datasets and list the metric change in Table 13 (Calabrese et al., 2019; Bi et al., 2020; Sironi et al., 2018; Orchard et al., 2015; Amir et al., 2017). ECNet denotes the network in which the Spatial-FA module of SECNet is replaced with a traditional convolution. The proportion of overall GMACs accounted for by the Spatial-FA module before and after the replacement is shown in Figure 6. Although SECNet generally underperforms ECNet across most datasets, it is markedly more efficient, with its GMACs constituting only 5% of the size of ECNet and its model parameters being smaller. Furthermore, SECNet has surpassed nearly all other representations in terms of performance.

## J. Look up Table for Runtime

Differences in the number of input events and the centroid of each loop in SECNet can influence its hardware performance. We evaluated the hardware performance of various combinations on the server platform, as shown in Table 12, measuring runtime in milliseconds (ms), parameters in millions (M), and MACs in giga (G). By observing the table, we can find that 1. the number of input events is independent of the model size, and the number of centroids per loop solely determines it; 2. The runtime increases synchronously with the event number and centroid. 3. GMAC super-linear grows with the increase of events number and centroid. Additionally, simply increasing the number of events will slow down the network inference, creating a trade-off between performance and inference speed in practical applications. The throughput of SECNet is guaranteed to be above 200 FPS when the centroid is kept at [512, 256, 128] and [256, 128, 64].

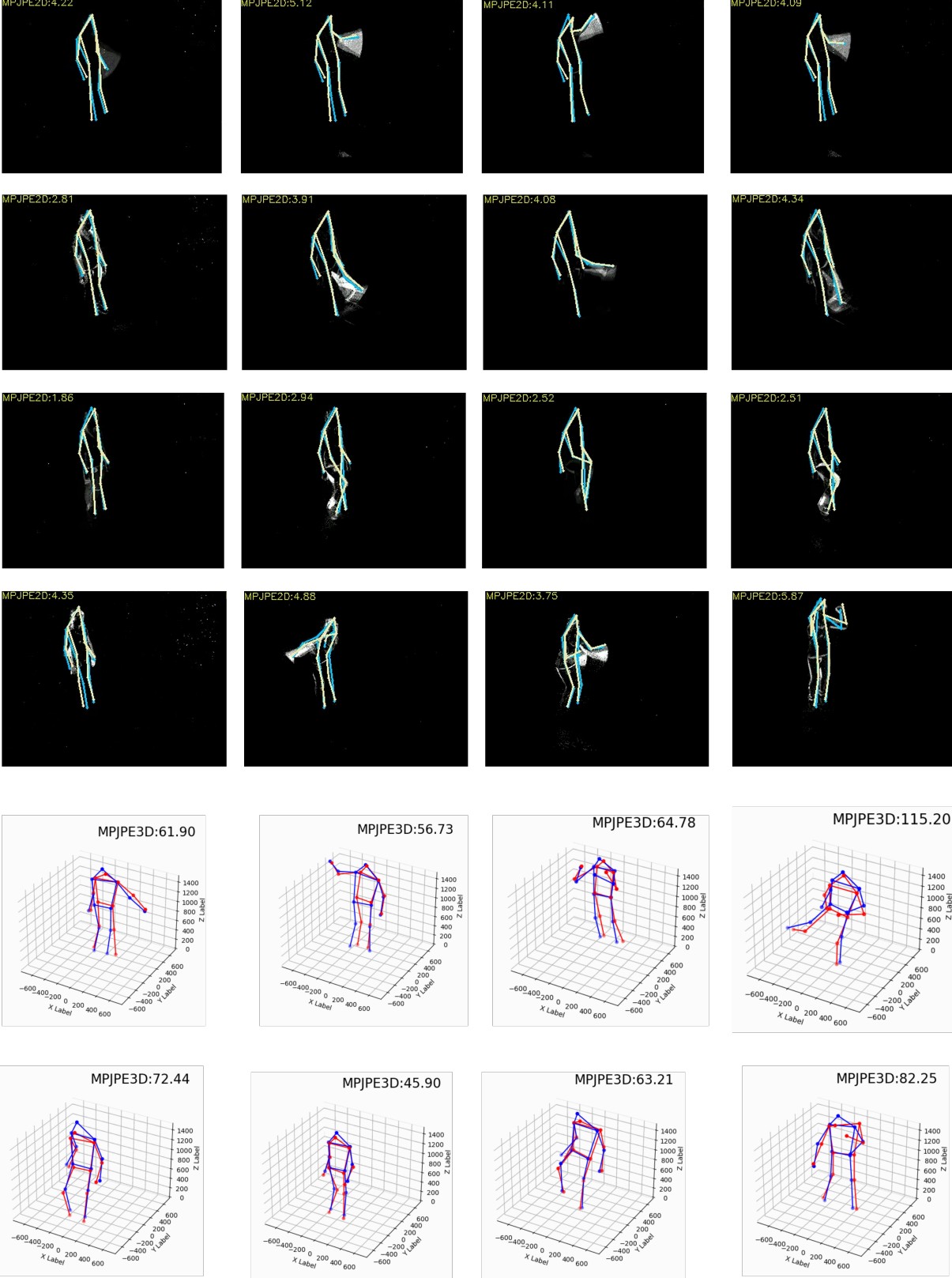

*Figure 8.* More 2D and 3D SECNet's results in DHP19 dataset.

| Dataset | DHP19 | | | | ASLDVS | | UCF101-DVS | | NCARS | | NMNIST | | DVS128 Gesture | |
|---|---|---|---|---|---|---|---|---|---|---|---|---|---|---|
| Method | SECNet | | ECNet | | SECNet | ECNet | SECNet | ECNet | SECNet | ECNet | SECNet | ECNet | SECNet | ECNet |
| Metric | 6.11 | 69.89 | 5.87 | 66.87 | 0.9993 | 0.9994 | 0.916 | 0.941 | 0.947 | 0.951 | 0.997 | 0.997 | 0.989 | 0.983 |
| Parameter | 1.831 | | 2.193 | | 1.107 | 1.469 | 1.126 | 1.489 | 0.896 | 1.258 | 1.103 | 1.465 | 0.898 | 1.26 |
| GMACs | 0.160 | | 0.904 | | 0.345 | 1.832 | 0.48 | 1.967 | 0.228 | 0.971 | 0.345 | 1.832 | 0.11 | 0.903 |
| SFA MMACs | 39.5 | | 783 | | 79 | 1566 | 79 | 1566 | 39.5 | 802 | 79 | 1566 | 39.5 | 783 |

*Table 13.* Ablation study for the Spatial-FA module of SECNet.

## K. Spatio-temporal Features

Event Cloud contains both spatial $(x, y)$ and temporal $t$ coordinates, and the features extracted through the network using these two types of coordinates refer to spatial-temporal features. Additionally, the manner in which these coordinates are employed results in distinct categories of features. We break down the categories of temporal features into **explicit** and **implicit** types. Previous works have treated the events' $t$ coordinate as the space $z$ coordinate, employing the Point Cloud network to abstract features where $t$ is regarded as an implicit variable analogous to $(x, y)$. So, we call such features implicit temporal features. In contrast, when $t$ is treated as an index within networks such as LSTM (Sherstinsky, 2020), SSM (Gu & Dao, 2023), and attention mechanisms (Niu et al., 2021), we classify the resulting features as explicit temporal features.

## L. Visualization

Figure 7 presents the prediction results of various models applied to the same test samples, whose settings are consistent with the paper (Chen et al., 2022a). ResNet18 and DHP19 employ the event frame representation (Calabrese et al., 2019), PointNet utilizes the rasterized Point Cloud representation (Chen et al., 2022a), and SECNet adopts the Event Cloud representation. The results of SECNet are superior to those of the other three methods. More results for different sequences in the dataset are shown in Figure 8.

