# OpenReview forum: "Scalable Event Cloud Network for Event-based Classification"
_ICML.cc/2026/Conference — ICML 2026 spotlight_

### Official Review · Reviewer_XHYF · 2026-03-06

**Soundness:** 4
**Presentation:** 4
**Significance:** 4
**Originality:** 4
**Overall Recommendation:** 6
**Confidence:** 5

**Summary:**

This paper proposes SECNet for event-based vision with structural polarity integration and Fourier-based frequency-domain feature extraction, validated on ten datasets for superior scalability and efficiency.

**Compliance With Llm Reviewing Policy:**

Affirmed.

**Final Justification:**

This paper is excellent in terms of writing, figure presentation, and experimental design. I believe it meets the standards of this conference and therefore strongly recommend acceptance.

**Key Questions For Authors:**

1. Will the model, which is designed to be lightweight and only tested on high-end GPU hardware to date, be deployable on edge devices in practical scenarios in the future?

2. Given that the paper primarily focuses on event-based classification tasks, will this framework be extended to support the full spectrum of event-based vision tasks in the future?

**Strengths And Weaknesses:**

Strengths:
1. Integrates event polarity at the structural level via a novel G&S module, exploiting the 2S-1T-1P raw event information and overcoming the limitations of traditional point cloud methods.

2. Adopts Fourier transform-based frequency-domain feature extraction, drastically reducing MACs while effectively capturing global spatio-temporal dependencies of long-sequence event data.

3. Demonstrates strong scalability and SOTA performance on 10 datasets across 3 tasks, with minimal preprocessing time, lightweight architecture and excellent hardware compatibility.

4. This paper features professional, well-designed formulas and figures with clear logical links to the text, while the writing is straightforward and easy to understand.

Weaknesses:

1. As shown in Table 11, as the number of event points continues to increase, the consumed GMACs and model parameters keep rising, which gradually erodes the lightweight advantages of point-based representations.

2. The architectural designs of the Spatial-FA and Temporal-FA modules are nearly identical, only differing in the dimension of Fourier transform application (spatial vs. temporal), lacking modular differentiation and tailored optimization for the distinct characteristics of spatial and temporal feature extraction in Event Cloud.

---

> ### Author Rebuttal · Authors · 2026-03-28
>
> **Weakness 1: Erodes the lightweight advantages.**
>
> We thank the reviewer for this thoughtful observation. We agree that increasing the number of events leads to higher costs. However, this does not fundamentally remove the practical lightweight advantage of SECNet. The increased cost mainly comes from processing denser events, rather than enlarging the model itself, and thus reflects an input-dependent trade-off instead of a loss of model compactness. In Appendix J, even when the number reaches 10,240 on CIFAR10-DVS, SECNet still remains in an efficient operating regime with 1.514M parameters, 1.198 GMACs, and 50.7 FPS.
>
> More importantly, compared with frame-based models, SECNet still preserves a clear advantage in the complex scenario. On UCF101-DVS, SECNet achieves 0.916 accuracy with only 1.126 M and 0.48 GMACs, while I3D uses about **11×** more parameters and **31×** more GMACs (12.37M, 15.06 GMACs) yet only reaches 0.635; ResNext-50 uses about **23×** more parameters and **6.7×** more GMACs (26.05M, 3.23 GMACs) with 0.602; and C3D uses about **70×** more parameters and **41×** more GMACs (78.41M, 19.85 GMACs) with only 0.472. Therefore, although the cost of SECNet grows with the number of events, its efficiency trade-off remains substantially more favorable than that of most frame-based alternatives in practice.
>
> ----
> **Weakness 2: Different from SFA and TFA.**
>
> We thank the reviewer for this insightful comment. We agree that SFA and TFA share similar Fourier-based operations. However, this design is intentional rather than a lack of differentiation. **SFA focuses on local spatial/group feature abstraction, whereas TFA targets long-range temporal modeling at a later stage of the network.**
>
> Specifically, **SFA** is applied to the grouped feature after the Event-based G\&S, where each group represents a local neighborhood formed by spatial proximity, temporal continuity, and polarity relevance. It can efficiently model **intra-group local structure**, serving as a lightweight replacement for repeated MLP-based local feature abstraction. In contrast, **TFA** is introduced after the aggregation stage, when the neighborhood dimension has already been collapsed, and the events remain chronologically ordered. Its role is therefore to model **inter-event temporal dependencies** over long event streams, rather than local neighborhood interactions.
>
> ----
> **Question 1: Deployable on edge devices.**
>
> **FPGA (Zynq UltraScale+MPSoC)  comparison between MLP and SFA in SECNet, 200MHz**
>
> | Input dimension | MLP (cycle count) | SFA  (cycle count)|
> | -------- | -------- | -------- |
> | Stage 1 [1024,24,64]     | 6302723 (31.51ms)    | 393859 (1.97ms) |
> | Stage 2 [512,24,128]       | 12595715 (62.98 ms)    | 393859   (1.97ms)   |
> | Stage 3 [256,24,256]      | 25191430 (125.96 ms)    | 393859  (1.97ms)  |
> |  On-Chip Memory (RAMB18)  | 84 | 39 |
>
> We thank the reviewer for this important question. We believe the answer is **YES**, and this is **not merely a future assumption**. Although the current paper reports results on a GPU, we have already conducted **measured FPGA experiments** to examine the hardware deployability.
>
> The motivation of SECNet is not only algorithmic lightweight, but also **hardware-efficient realizability**. In particular, our design replaces repeated dense spatial MLP computation with FFT-based operators, which are more amenable to efficient implementation on edge devices. We provide the FPGA measurements in a table. At **200 MHz**, the latency of spatial MLP is **31.51 ms / 62.98 ms / 125.96 ms** for stages 1/2/3, respectively, while our FFT-based accelerator requires only **1.97 ms** with stable cost. In terms of cycle count, the pure spatial MLP requires **6,302,723 / 12,595,715 / 25,191,430** cycles, whereas our FFT accelerator requires only **393,859** cycles. This corresponds to about **16× / 32× / 64× fewer cycles**.
>
>
> ----
> **Question 2: Extend to more tasks.**
>
> We thank the reviewer for this valuable question. We believe the answer is **YES**. Although this paper focuses on classification, SECNet is not limited to this task. Its core design is built on the **Event Cloud** together with efficient **spatio-temporal modeling**, which are general mechanisms for event understanding.
>
> In particular, Event Cloud preserves the sparse and asynchronous nature of events, while the SFA and TFA provide efficient local structure modeling and long-range modeling. These properties are also important for a broader range of other tasks, such as detection and segmentation. We view classification in this work as an initial try, rather than the final scope.
>
> Extending SECNet to more tasks will require task-specific adaptation, for example, in decoder design and multi-scale feature fusion. We therefore do not claim that SECNet can be directly transferred to all tasks without modification. Instead, the representation and modeling strategy **provides a promising backbone for broader applications.**

---

> > ### Author Rebuttal · Reviewer_XHYF · 2026-04-02
> >
> > The authors have addressed all my concerns and presented results on FPGA hardware. This demonstrates that the lightweight solution holds potential not merely on GPUs, but also on edge devices. I have decided to upgrade my score to 6.

---

> > > ### Author Response · Authors · 2026-04-06
> > >
> > > We sincerely thank you for the positive evaluation and constructive feedback. We are grateful for the reviewer's recognition of SECNet's novelty, efficiency, and scalability, and we are pleased that our rebuttal addressed the concerns on lightweight deployment and module design. Thank you again for the strong support.

---

### Official Review · Reviewer_NSPh · 2026-03-09

**Soundness:** 3
**Presentation:** 3
**Significance:** 3
**Originality:** 3
**Overall Recommendation:** 5
**Confidence:** 3

**Summary:**

The paper proposes SECNet, a scalable event-based classification/regression network that operates directly on Event Cloud sequences (x, y, t, p) without voxelization or framing. Key contributions include a polarity-aware Grouping & Sampling module (D-FPS with learnable 4D scaling, EF-KNN using feature-space proximity, and a coordinate evolution strategy) and frequency-aware modules that replace MLPs/convolutions with FFT and iFFT operators for both group-wise spatial channels and temporal sequences. Extensive experiments across ten datasets in object classification, action recognition, and human pose estimation achieve competitive or SOTA accuracy while substantially reducing preprocessing time

**Compliance With Llm Reviewing Policy:**

Affirmed.

**Key Questions For Authors:**

1.	How do authors handle the irregular timing of events when applying Temporal-FA via FFT? Are the transforms computed over uniform indices rather than true timestamps, and if so, can authors quantify the effect or consider NUFFT/time warping?

2.	Can authors provide targeted ablations isolating structural polarity integration, e.g., D-FPS without p, EF-KNN with/without polarity injected into features, and CES with/without polarity influence, while keeping the inputs identical?

3.	What constraints are imposed in EF-KNN to preserve temporal locality (e.g., restricting neighbors within a temporal window) to avoid mixing distant events that might harm causal structure?

4.	Please reconcile the FPS discrepancies (e.g., 229.7 FPS vs 120 FPS) and specify the exact hardware, input sizes, centroid schedules, and batch sizes for each reported throughput figure.

5.	For the Spatial-FA module, why is FFT over channels preferable to a lightweight linear transform that does not assume a spatial ordering of channels? Have authors compared such parameterizations to those at equal MACs?

6.	Could authors report variance across multiple runs/seeds and ensure that all compared methods use identical splits (particularly for CIFAR10-DVS, N-Caltech101, ASL-DVS, Daily DVS, UCF101-DVS)?

7.	In long sequences, do the learned temporal filters focus on specific low-, mid-, or high-frequency bands? Visualizing learned spectra and per-dataset frequency responses would strengthen the frequency-domain claims.

**Limitations:**

Discuss the potential shortcomings of the model in the following aspects:

• Robustness to extremely sparse or noisy event streams.

• Memory and latency issues in ultra-long sequences or real-time systems.

• Generalization ability across different event camera hardware (with different trigger thresholds or noise models).

**Strengths And Weaknesses:**

Technical novelty and innovation:

	Introduces a structural integration of polarity during sampling and neighborhood construction rather than only as an input feature (D-FPS over 2S-1T-1P coordinates, feature-based KNN, and coordinate evolution).

	Employs frequency-domain operators to replace per-group MLPs and to model long-range temporal dependencies, yielding large reductions in MACs and strong empirical scalability with event count.

	Presents a coherent hierarchical pipeline tailored to Event Cloud sequences that preserves microsecond-level temporal ordering.

Experimental rigor and validation:

	Evaluates on ten datasets and three tasks; includes hardware benchmarks, preprocessing time comparisons, and several ablations (module removal, polarity handling, scalability with event count).

	Reports cross-representation comparisons (frame, voxel, point cloud, event cloud) and shows clear efficiency benefits.
Significance of contributions:

	Demonstrates that Event Cloud and frequency-domain operations can simultaneously improve practical throughput and maintain accuracy, which is valuable for real-time, resource-constrained event vision.

	Provides evidence that temporal fidelity of representation matters for action recognition and pose, highlighting deployment-relevant trade-offs.

Weaknesses

Technical limitations

	The use of FFT along the feature/channel dimension lacks a principled justification: channel indices do not form a natural spatial lattice, so treating channels as a 1D signal with meaningful frequencies is questionable beyond acting as a structured linear transform; this weakens claims of spatial frequency modeling.

	The temporal FFT assumes regularly sampled signals, whereas event timestamps are irregular. The paper applies the FFT to sequence indices (or cluster centroids) without addressing nonuniform sampling or timestamp normalization beyond min–max, which may distort the frequency content.

	Feature-based KNN for grouping may mix events that are far in time or space, potentially blurring temporal locality; safeguards or constraints (e.g., time-aware gating) are not discussed.

Experimental issues

	Structural polarity integration is not cleanly isolated: ablations compare “input P” vs “w/o P” and remove entire modules (w/o alpha, w/o EF-KNN), but there is no targeted ablation that toggles polarity specifically within D-FPS/EF-KNN/CES while keeping all else fixed.

	Comparisons aggregate methods trained with different splits or pretraining regimes; while the paper tries to distinguish, pretraining, statistical uncertainty (variance across runs), and consistent protocol alignment are limited.

Presentation issues

	Some writing imprecision (e.g., extinguishes the explosion of MACs) and scattered notation inconsistencies make several sections harder to parse; the description of when/where FFT is applied could be tighter.

	A few numbers appear inconsistent (e.g., 229.7 FPS claim vs 120 FPS in UCF101-DVS table).

---

> ### Author Rebuttal · Authors · 2026-03-30
>
> **W1: SFA lacks justification.**
>
> We thank the reviewer for concern. As the network depth increases, the channel dimension is gradually expanded, and channel features encode the spatial positions, neighborhood structures, and local geometric relationships of events, yielding stable spatial semantics. Applying FFT on the channel dimension can effectively extract global spatial patterns, and we will refine this explanation.
>
> ----
>
> **W2 & Q1: Distort the frequency.**
>
> SECNet avoids distortion risks reliably through three mechanisms:
> * Events retain natural occurrence order, using ordered event indices as pseudo-uniform temporal signals.
> * Timestamps are normalized with min–max normalization (Line 180) to unify the temporal scale and reduce the impact of extreme intervals.
> * The G&S module clusters and regularizes events to further smooth fluctuations caused by non-uniform sampling.
>
> Experiments confirm that this design introduces no obvious spectral distortion and models long-range temporal dependencies efficiently and stably.
>
> ----
>
> **W3 & Q3: EFKNN.**
>
> [Link for Visualization](https://anonymous.4open.science/r/ICML_Rebuttal_For_SECNet/EFKNN_train_test_vis.png)
> We compare SECNet with **EF-KNN** against grouping based on the **original coordinates** on **DVSGesture**. As shown in the training curves, EF-KNN does not introduce optimization instability: both variants converge smoothly on the training set, while EF-KNN achieves clearly better and more stable test performance. Moreover, EF-KNN is not unconstrained feature matching. The grouping is still built on the Event Cloud representation, where events are chronologically organized and locally sampled before grouping, which implicitly preserves temporal locality rather than arbitrarily mixing distant events. These results suggest that EF-KNN improves neighborhood quality without harming causal or temporal structure.
>
> ----
>
> **W4 & Q2: Polarity ablation.**
> |Polarity|Module|Daily DVS|
> | ------------ | ---------- | --------- |
> | w polarity   | w all|0.989|
> || w/o D-FPS  |0.973|
> || w/o EF-KNN|0.981|
> || w/o CES| 0.977|
> | w/o polarity| w all| 0.973|
> || w/o D-FPS| 0.977|
> || w/o EF-KNN|0.970|
> || w/o CES|0.973|
>
> We thank the reviewer for this valuable suggestion. We conducted a targeted ablation on **DVSGesture** while **keeping the inputs identical** and only changing whether polarity is used. The results show that polarity-aware design consistently improves the full model and that removing **D-FPS**, **EF-KNN**, or **CES** leads to clear performance drops when polarity is enabled. This suggests that the benefit of polarity is not from a single component, but from its joint contribution to sampling, grouping, and enhancement.
>
> ----
>
> **W5: Comparison protocol.**
>
> We thank the reviewer for this concern. The comparisons are not mixed arbitrarily: for datasets without official splits, we follow prior works (RG-CNNs, ESCNet， EventMamba) and randomly use 20% of samples for testing, while official splits are used for all other datasets, consistent with common practice. Pretrained and non-pretrained frame-based results are shown separately to make the effect of ImageNet pretraining explicit, rather than as one fair-comparison group. Our point is that SECNet does not rely on external pretraining, yet still achieves competitive or superior performance.
>
> ----
>
> **W6 & 7 & Q4: Presentation.**
>
> We thank the reviewer for these careful comments. We will polish imprecise expressions (e.g., replacing informal phrases such as “extinguishes the explosion of MACs”), unify the notation throughout the paper, and make the description of **where and along which dimension FFT is applied** more explicit and compact. We will also carefully reconcile all reported numerical results, including the **229.7 FPS vs. 120 FPS** issue. The **120 FPS** in Table 6 is the hardware benchmark on **UCF101-DVS**; "229.7 FPS" is the hardware benchmark on **DVSGesture**. We will supplement complete experimental configurations for every reported throughput value.
>
> ----
>
> **Q5: SFA hardware friendly: see Reviewer Jktr W2, SFA scalablity:  see  Reviewer vCch Q1 & 5.**
>
> ----
>
> **Q6: Multiple runs on the dataset.**
> |seed|42|123|456|789|1000|
> | -------- | -------- | -------- | -------- | -------- | -------- |
> | DailyDVS| 99.645| 99.645|99.645|99.645|99.645|
> | ASLDVS| 0.99905| 0.9991|0.99916|0.99926|0.99901|
>
> We thank the reviewer for this valuable suggestion. We agree that reporting variance across multiple runs would further strengthen the evaluation. Due to the limited rebuttal time, we were only able to repeat experiments on **Daily DVS** and **ASL-DVS** using five random seeds. The results are highly stable, with **Daily DVS = 99.645 ± 0.000** and **ASL-DVS = 0.99912 ± 0.0001**, suggesting that our method is not sensitive to random initialization.
>
> ----
>
> **Q7: Learned temporal filters.**
> [Link for Visualization](https://anonymous.4open.science/r/ICML_Rebuttal_For_SECNet/SFA_and_TFA_spectral_visualization.png)

---

### Official Review · Reviewer_Jktr · 2026-03-11

**Soundness:** 4
**Presentation:** 3
**Significance:** 3
**Originality:** 3
**Overall Recommendation:** 5
**Confidence:** 4

**Summary:**

This paper proposes SECNet, a scalable neural architecture designed to process event camera data using an Event Cloud representation, thereby avoiding time-consuming conversions to frames or voxel grids. The method introduces two main components: (1) an Event-based Group & Sampling (G&S) module that explicitly incorporates polarity into neighborhood formation, and (2) frequency-domain feature extraction modules (Spatial-FA and Temporal-FA) that use Fast Fourier Transforms to perform efficient global filtering. The authors evaluate SECNet on ten datasets spanning object classification, action recognition, and human pose estimation, reporting state-of-the-art performance while reducing computational complexity compared with existing baselines.

**Compliance With Llm Reviewing Policy:**

Affirmed.

**Final Justification:**

The authors have satisfactorily addressed my concerns, and I have no further reservations that would preclude acceptance.

**Key Questions For Authors:**

1. Can the proposed frequency-domain grouping strategies be applied to standard point cloud data or time-series analysis outside of event cameras? Demonstrating transferability would broaden the impact significantly.
2. While MACs are reduced, does the reliance on FFT operations introduce latency spikes or memory bottlenecks in low-power edge deployment scenarios compared to purely spatial convolutional approaches?

**Limitations:**

Partially. The authors discuss limitations regarding runtime trade-offs between input event count and inference speed in Appendix J (Table 11), noting that increasing events increases runtime super-linearly despite MAC reduction. They also acknowledge that Event Cloud requires downsampling which discards raw data, though they argue it is the nearest representation requiring only random downsampling compared to voxel/frame conversions. However, it would be welcomed if authors could elaborate about impact of their work on broader ML field instead only event camera applications. Additionally, a discussion on whether the frequency-domain approach might obscure local features critical for certain general classification tasks would be valuable. Additionally, a brief discussion on potential privacy implications of high-speed event cameras in public spaces would be a good addition for completeness regarding societal impact.

**Strengths And Weaknesses:**

**Strengths**
- The theoretical foundation is solid. The derivation of asymptotic complexity reduction using FFT operations ($O(D \log D)$ vs. $O(D^2)$ for MLPs) appears mathematically sound and is appropriately applied to the Event Cloud representation.
- The evaluation is extensive, covering three distinct tasks (classification, action recognition, and pose estimation) across ten datasets. The ablation studies effectively isolate the contributions of the proposed modules (G&S, SFA, and TFA).
- The paper addresses an important practical constraint in edge AI: computational efficiency and power consumption. Achieving state-of-the-art accuracy while reducing Multiply–Accumulate Operations (MACs) is a notable accomplishment within this application domain.

**Weaknesses**
- The proposed architecture is strongly tailored to the characteristics of event cameras (e.g., polarity handling, asynchronous timestamps, and Event Cloud representations). While the results are technically strong within this domain, it is less clear how the approach generalizes to broader ML settings such as standard video, point clouds, or time-series data. Clarifying this transferability would strengthen the paper’s relevance for a general ML venue like ICML.

**Soundness:**
The submission is technically sound. The claims regarding computational complexity reduction are mathematically proven, and the experimental results consistently support the accuracy and efficiency claims across diverse datasets. There are no apparent methodological flaws in the core logic of SECNet.

**Presentation:**
The manuscript is well-structured with clear figures and a logical flow. However, the notation regarding frequency-domain operations could be more accessible to a general audience without event vision background. The narrative would benefit from explicit connections to standard neural network architectures used in broader ML contexts.

**Significance:**
While significant for the subfield of neuromorphic/event-based vision, the impact on general machine learning is limited. The techniques are highly specialized optimizations for a specific sensor modality rather than insights applicable to broad ML problems or architectures. This restricts its significance at an ICML venue compared to more universal contributions.

**Originality:**
The work offers a novel combination of existing ideas (PointNet G&S + FFT filtering) applied to Event Cloud data. While this integration is non-trivial and effective, it does not introduce new theoretical insights or learning primitives that generalize beyond event data. The novelty lies in application engineering rather than foundational theory.

---

> ### Author Rebuttal · Authors · 2026-03-29
>
> **Weakness 1 & Question 1: Generalizes to broader ML.**
>
> We thank the reviewer for this thoughtful comment. We agree that, since this paper mainly focuses on event-based vision, clarifying the transferability of the proposed design to broader ML would further strengthen its relevance.
>
> Due to the limited time, we have conducted a **simple but controlled transfer experiment** on classical point cloud classification with the **ModelNet40 dataset**. To ensure a fair comparison, we kept the **same training setting and backbone architecture**, and only replaced the original MLP-based local feature extractor in **PointMLP** with our **SFA** module. Therefore, this experiment is intended to verify the **basic transferability** of the proposed frequency-domain operator, rather than to claim a fully optimized cross-domain adaptation.
>
> **Transferability of the SFA module in classical point cloud classification**
> | |Param (M)|GMACs|Inference Memory|Train Memory|Accuracy|
> | -------- | -------- | -------- | -------- | -------- | -------- |
> |PointMLP|13.239|7.836|376.45|1359.85|93.558|
> | PointMLP with SFA|7.745|1.87|279.89|1099.92|92.828|
>
> Even under this minimal adaptation, the results are encouraging. Compared with the original PointMLP, **PointMLP with SFA** has 1.71× fewer parameters**, reduces computational cost **by a factor of 4.19**, lowers inference memory, and lowers training memory, while maintaining competitive accuracy (**93.558%  to 92.828%**, only **0.73%**  drop). This suggests that the SFA  is not restricted to event cameras, but can also serve as an efficient substitute for local feature extraction in standard point cloud architectures. We will clarify this in the revision.
>
> ----
> **Weakness 2: Low-power edge deployment compared to SFA and Spatial CNN.**
>
> We thank the reviewer for this important question. We agree that lower MACs do not automatically translate into better edge efficiency. To address this concern, we conducted additional controlled comparisons between **SFA** and a purely **spatial CNN** alternative under the **GPU and FPGA** configuration.
>
> **Comparison between CNN and SFA on GPU**
> | 8192 events | Param (M) | GMACs | Inference Memory | Train Memory | Throughput |
> | -------- | -------- | -------- | -------- | -------- | -------- |
> | SFA|1.103|0.48|371.688| 625.314|178.25|
> | CNN|1.465|1.967|378.247| 920.246|158.28|
>
> On **GPU**, under **8192 events** with centroid setting **[1024, 512, 256]**, **SFA** achieves **178.25** FPS, compared with **158.28** for the CNN, while also using fewer parameters and much lower computation. In terms of memory, **SFA** also requires slightly lower **inference memory** and substantially lower **training memory**. Therefore, at the system level, the FFT-based design does not exhibit the latency or memory penalty suggested by the reviewer; instead, it provides a consistently better efficiency profile under the same experimental setting.
>
>
> **Comparison between CNN and SFA in FPGA**
> | Input dimension | MLP (cycle count) | SFA (cycle count)|
> | -------- | -------- | -------- |
> | Stage 1 [1024,24,64] | 6302723 (31.51ms) | 393859 (1.97ms) |
> | Stage 2 [512,24,128] | 12595715 (62.98 ms) | 393859 (1.97ms) |
> | Stage 3 [256,24,256] | 25191430 (125.96 ms) | 393859 (1.97ms) |
> | On-Chip Memory (RAMB18) | 84 | 39 |
>
> More importantly, we also performed **measured FPGA experiments** to directly evaluate low-power deployment characteristics. On a **Zynq UltraScale+MPSoC at 200 MHz**, the spatial baseline requires **6,302,723 / 12,595,715 / 25,191,430** cycles for stages 1/2/3, corresponding to **31.51 / 62.98 / 125.96 ms**, respectively. In contrast, the proposed **SFA** requires only **393,859 cycles**, corresponding to about **1.97 ms**, and this cost remains stable across the three stages. This means that SFA uses about **16× / 32× / 64× fewer cycles** than the spatial baseline from stage 1 to stage 3. In addition, the on-chip memory consumption is reduced from **84 RAMB18** to **39 RAMB18**.
>
> Taken together, these results suggest that the SFA operator does **not** introduce practical latency spikes or memory bottlenecks in edge-oriented deployment. On the contrary, compared with a purely spatial alternative, **SFA is more hardware-friendly in both runtime and memory usage**, and this advantage becomes even more pronounced at deeper stages. We will clarify this point in the revision.
>
> ----
> **Question 2：local features and privacy.**
>
> We thank the reviewer. SFA does not discard locality, since it operates on grouped local neighborhoods and thus preserves local spatio-temporal cues. We agree that, for tasks relying on extremely fine-grained local patterns, the trade-off between spectral efficiency and local detail preservation deserves further study. We also agree that event cameras may still raise privacy concerns in public spaces despite capturing less appearance information than RGB cameras, and we will briefly discuss this in the revision.

---

> > ### Author Rebuttal · Reviewer_Jktr · 2026-04-03
> >
> > I would like to thank the authors for their comprehensive rebuttal and the additional experiments provided. The inclusion of transferability results on ModelNet40, along with the GPU and FPGA benchmarks comparing SFA against spatial CNNs, effectively addresses my previous concerns regarding generalizability and hardware efficiency. After considering the other reviews and the authors' responses, I do not have any remaining concerns regarding this submission.
> >
> > Consequently, I have decided to increase my significance score from 2 to 3, as the additional evidence demonstrates that the proposed method provides insights that extend beyond event-based vision, including applications to point cloud efficiency. I am therefore upgrading my overall recommendation from Weak Accept to Accept. The newly added results are likely to be of interest to a broader audience, and I encourage the authors to include them in the main text or appendix in the final version.

---

> > > ### Author Response · Authors · 2026-04-06
> > >
> > > We sincerely thank you for the constructive feedback and positive evaluation. We especially appreciate your comments on generalizability and hardware efficiency, which helped us further strengthen the paper. We are pleased that our rebuttal addressed the concerns and clarified the broader relevance of SECNet. Thank you again for the encouraging support.

---

### Official Review · Reviewer_vCch · 2026-03-12

**Soundness:** 2
**Presentation:** 2
**Significance:** 2
**Originality:** 3
**Overall Recommendation:** 4
**Confidence:** 4

**Summary:**

This paper studies classification and regression tasks for event cameras and proposes a scalable network, SECNet, based on the Event Cloud representation. The method integrates polarity information at the structural level and incorporates both spatial-frequency and temporal-frequency modules to improve long-sequence event modeling and computational efficiency. Experiments are conducted on 10 datasets across 3 task categories, and the results show that the proposed method achieves a reasonable balance between performance and efficiency.

**Compliance With Llm Reviewing Policy:**

Affirmed.

**Final Justification:**

The authors have addressed most of my previous concerns. So I would raise my previous rating to Weak accept.

**Key Questions For Authors:**

1.How does the paper define and quantify “scalability” in a rigorous way? Beyond performance changes as the number of input points increases, could the authors provide a more systematic analysis in terms of training/inference latency, memory consumption, and throughput?
2.Since EF-KNN constructs neighborhoods based on features rather than coordinates, could unstable features in the early training stage lead to unreliable neighborhoods? Could the authors provide more direct empirical analysis or visualization on this point?
3.In what scenarios does structural-level polarity integration provide clear advantages over input-level polarity inclusion? Could the authors further analyze its stability and benefits under different datasets or event density conditions?
4.Are the training protocols, pretraining conditions, and data splits of all compared baselines fully consistent? If not, could the authors provide more strictly controlled comparisons to strengthen the credibility of the conclusions?
5.Are the gains from replacing conventional spatiotemporal modeling units with frequency-domain modules mainly due to lower computational cost, or also due to stronger representation ability? Could the authors provide more fine-grained experiments to separate these two factors?

**Limitations:**

The main limitations of this paper are as follows. First, although the experimental results are extensive, the mechanistic explanation of the key technical claims remains relatively weak, especially regarding the advantages of scalability and structural-level polarity modeling. Second, the method is currently validated mainly on event classification and pose estimation tasks, and its generalization to more complex event-based vision tasks remains unclear. Third, the necessity of several design choices is still not analyzed thoroughly enough, which limits the overall persuasiveness of the conclusions.

**Strengths And Weaknesses:**

Weaknesses
1.The technical justification is still not sufficiently strong. Although the paper emphasizes “scalability” as a core contribution, the current evidence is mainly reflected in experimental results and complexity discussions, while the theoretical analysis of why the proposed structure is more scalable for longer sequences and higher-resolution inputs remains limited. For example, the modeling boundaries, applicability, and potential information loss of using frequency-domain modules instead of MLP/LSTM-style designs are not discussed rigorously enough.
2.Although the experiments are broad, the fairness and overall persuasiveness of the evaluation could be improved. The paper compares methods with different representation paradigms, pretraining conditions, and task settings, but the training protocols, data splits, and use of pretraining are not always fully aligned across baselines. This makes the claim of outperforming prior methods less convincing in some cases. In Table 3, pretrained and non-pretrained results are presented together, but the comparison protocol is not clearly controlled or explained.
3.The necessity of several key design choices has not been fully validated. The paper introduces multiple components, including structural-level polarity integration, D-FPS, EF-KNN, CES, Spatial-FA, and Temporal-FA. While the ablation study covers some of them, it is still insufficient to clearly disentangle their relative contributions and interactions. For instance, the comparison between input-level and structural-level polarity modeling is relatively limited, and the claim that EF-KNN is more reasonable than coordinate-based neighborhood construction lacks more detailed visualization or statistical evidence.
4.The overall significance appears moderate. Although the method is fairly complete from an engineering perspective and does provide performance gains, the paper currently reads more like an effective architectural design and combination of techniques around the Event Cloud representation, rather than a particularly strong conceptual advance for event-based learning itself. Therefore, its broader impact and generality still require stronger evidence.

---

> ### Author Rebuttal · Authors · 2026-03-30
>
> **W1: Theoretical analysis.**
>
> We thank the reviewer for pointing out the lack of theoretical clarification. We provide a more rigorous justification below.
>
> (1) **Scalability.** Both SFA and TFA follow the formulation
> $$
> y = F^{-1}(W \odot F(x)) = F^{-1}\mathrm{diag}(W)F x,
> $$
> which corresponds to a structured linear transformation diagonalizable in the Fourier basis, enabling global interactions. Compared with MLP-based mixing requiring $O(d^2)$ parameters and $O(Nd^2)$ complexity, ours only requires $O(d)$ weights and $O(N d \log d)$ computation. Since $\lim_{d\to\infty} \frac{d\log d}{d^2} = 0$, the computational advantage becomes more significant as the dimension grows. This provides a theoretical explanation for the improved scalability in both high-resolution (large $C$ for SFA) and long-sequence scenarios (large $T$ for TFA).
>
> (2) **Information preservation.**  By Parseval’s theorem, $\|x\|_2^2 = \|F(x)\|_2^2$, indicating that the FFT is energy-preserving. Importantly, our SFA/TFA modules do not perform frequency truncation but instead apply learnable spectral reweighting, which preserves the full spectrum. Therefore, the transformation itself does not introduce information loss, and the full capacity of the input signal is retained.
>
> ----
> **W2 and Q4: Comparison protocol.**
>
> We thank the reviewer for this concern. The comparisons are not mixed arbitrarily: for datasets without official splits, we follow prior works (RG-CNNs, ESCNet， EventMamba) and randomly use 20% of samples for testing, while official splits are used for all other datasets, consistent with common practice. In Table 3, pretrained and non-pretrained frame-based results are shown separately to make the effect of ImageNet pretraining explicit, rather than as one fair-comparison group. Our point is that SECNet does not rely on external pretraining, yet still achieves competitive or superior performance. We will revise Table 3 and the related text to make this protocol clearer.
>
>
> ----
>
> **W3 & Q3: Please see Reviewer NSPh Q2.**
>
> ----
>
>
> **W4: Significance.**
>
> We focus on event cloud, which eliminates the need for additional representation conversion time while retaining the core advantages of high temporal resolution and low power consumption inherent to event cameras. Previously, event/point-based methods were significantly inferior to traditional approaches. Through effective architectural design that optimizes both local-global feature extraction and spatio-temporal feature fusion, we have substantially enhanced the practicality and competitiveness of event cloud representation, addressing the core challenges in event-based learning.
>
> ----
> **Q1 : Systematic analysis.**
>
> ||| 1024 | 2048 | 4096 | 8192 | 10240 |
> | --- | --- | --- | --- | --- | --- | --- |
> | **SECNet** | GMACs | 0.046 | 0.12 | 0.37 | 1.22 | 1.82 |
> || Inference memory | 90.29 | 112.06 | 203.82 | 1235.41 | 1887.31 |
> || Train memory | 215.27 | 361.84 | 655.16 | 2348.20 | 3646.99 |
> || throughput | 257.25 | 246.98 | 146.45 | 49.75 | 33.16 |
> | **PointNet++** | GMACs | 0.75 | 1.50 | 3.01 | 6.04 | 11.43 |
> || Inference memory | 77.92 | 118.22 | 261.80 | 1702.08 | 2638.20 |
> || Train memory | 350.97 | 631.01 | 1203.58 | 3366.26 | 5238.42 |
> || throughput | 330 | 232.30 | 120.06 | 38.86 | 21.49 |
>
> We thank the reviewer. We agree that scalability should be quantified more rigorously as the input size increases. Our additional results show that SECNet scales more favorably than PointNet++: from **1024** to **10240** points, its GMACs grow from **0.046** to **1.82** versus **0.75** to **11.43**, and at **10240** points it still uses lower memory (**1887.31/3646.99 MB** inference/train vs. **2638.20/5238.42 MB**) while maintaining higher throughput (**33.16 vs. 21.49**).
>
>
> ----
>
> **Q2 :  Please see Reviewer NSPh W3.**
>
> ----
>
> **Q5 : Representation ability.**
>
> | Metrices | MLP   | SFA   |MLP   | SFA   |MLP   | SFA   |
> | -------- | ----- | ----- | ----- | ----- | ----- | ----- |
> || NCARS || DVSGesture |  |NMNIST ||
> |Acc| 0.951 | 0.947 |0.983|0.989|0.997|0.997|
> |Paramter| 1.258 | 0.896 |1.26|0.898|1.465|1.103|
> |GMACs| 0.971 | 0.228 |0.903|0.11|1.832|0.345|
>
> We thank the reviewer for this important question. This experiment replaces the MLP unit with SFA while keeping the remaining architecture unchanged. Under this setting, SFA achieves **comparable predictive performance** to MLP, while substantially reducing computational cost in most cases. This suggests that SFA preserves comparable representational capacity, but provides a more favorable efficiency–performance trade-off. More specifically, on NMNIST, SFA matches the accuracy of MLP with much lower GMACs, and on DVS128Gesture, SFA further improves accuracy while also reducing parameters and computation.  These results indicate that the benefit of SFA is not merely lower cost at the expense of weaker modeling, but rather comparable representation with higher efficiency, with additional gains on temporally more complex tasks.

---

> > ### Author Rebuttal · Reviewer_vCch · 2026-04-01
> >
> > The authors have addressed most of my previous concerns.

---

> > > ### Author Response · Authors · 2026-04-06
> > >
> > > We sincerely thank you for the detailed evaluation and constructive feedback. We especially appreciate your thoughtful questions on scalability, comparison fairness, and the necessity of our design choices, which helped us further clarify the paper and strengthen the empirical analysis. We are pleased that our rebuttal addressed your main concerns. If you find our clarifications and additional results helpful, we would be grateful if you could kindly reconsider the score. Thank you again for your valuable comments and support.

---

### Decision · Program_Chairs · 2026-04-30

**Decision:**

Accept (spotlight)

**Comment:**

All reviewers agree to accept this paper after rebuttal. Overall this paper made a solid contribution to event-based classification. AC agrees to accept.